# Ferrari: Federated Feature Unlearning via Optimizing Feature Sensitivity

Hanlin Gu[2,*],    Win Kent Ong[1*],    Chee Seng Chan[1†], and    Lixin Fan[2]

[1]CISiP, Universiti Malaya, Malaysia
[2]AI Lab, Webank, PR China

## Abstract

The advent of Federated Learning (FL) highlights the practical necessity for the *'right to be forgotten'* for all clients, allowing them to request data deletion from the machine learning model's service provider. This necessity has spurred a growing demand for Federated Unlearning (FU). Feature unlearning has gained considerable attention due to its applications in unlearning sensitive, backdoor, and biased features. Existing methods employ the influence function to achieve feature unlearning, which is impractical for FL as it necessitates the participation of other clients, *if not all*, in the unlearning process. Furthermore, current research lacks an evaluation of the effectiveness of feature unlearning. To address these limitations, we define feature sensitivity in evaluating feature unlearning according to Lipschitz continuity. This metric characterizes the model output's rate of change or sensitivity to perturbations in the input feature. We then propose an effective federated feature unlearning framework called Ferrari, which minimizes feature sensitivity. Extensive experimental results and theoretical analysis demonstrate the effectiveness of Ferrari across various feature unlearning scenarios, including sensitive, backdoor, and biased features. The code is publicly available at https://github.com/OngWinKent/Federated-Feature-Unlearning

## 1 Introduction

Federated Learning (FL) [1–3] allows for model training across decentralized devices or servers holding local private data samples, without the need to exchange them directly. An essential requirement within FL is the participants *"right to be forgotten"*, as explicitly outlined in regulations such as the European Union General Data Protection Regulation (GDPR)[3] and the California Consumer Privacy Act (CCPA)[4] [4]. To address this requirement, Federated Unlearning (FU) has been introduced, enabling clients to selectively remove the influence of specific subsets of their data from a trained FL model while preserving the model's accuracy on the remaining data [5].

Different from unlearning at the *client, class, or sample* level [6–8] in FL, the feature unlearning [9] holds significant applications across various scenarios. Firstly, in contexts where sentences contain sensitive information such as names and addresses [9, 10], it becomes crucial to remove these sensitive components to prevent potential exposure through model inversion attacks [11–14]. Secondly, when datasets contain backdoor triggers that can compromise model integrity [15–18], it is imperative to eliminate these patterns. Thirdly, unlearning biased features becomes essential in scenarios where data imbalances significantly impact model accuracy due to bias [19–22]. However,

---

[*]equal contribution; authors are listed alphabetically by first name.

[†]corresponding author (*cs.chan@um.edu.my*).

[3]`https://gdpr-info.eu/art-17-gdpr/`

[4]`https://oag.ca.gov/privacy/ccpa`

existing works of FU focus on client, class, or sample unlearning [6–8] but do not address feature unlearning, limiting their ability to unlearn specific features across multiple data points.

There are two challenges in feature unlearning in FL. Firstly, evaluating the unlearning effectiveness for feature unlearning is difficult. Typically, unlearning effectiveness is assessed by comparing the unlearned model with a retrained model without the feature. However, building data without the feature is challenging; for example, training the data with noise or a black block on the feature region may cause severe degradation in model accuracy (see Sec. 3.2). Secondly, previous work on feature unlearning within centralized machine learning settings [9, 10] is not practical for FL due to its requirement for access to all datasets, necessitating the participation of all clients.

To address the aforementioned limitations, we first define the feature sensitivity in Sec. 4.1 to evaluate the feature unlearning inspired by the Lipschitz continuity, which characterizes the rate of change or sensitivity of the model output to perturbations in the input feature. Then we propose a simple but effective federated feature unlearning method, called Ferrari (**Fed**erated Feature Unlearning), by minimizing the feature sensitivity in Sec. 4.2. Our Ferrari framework offers three key advantages: Firstly, Ferrari requires only local datasets from the unlearned clients for feature unlearning. Secondly, Ferrari demonstrates high practicality and efficiency, which support various feature unlearning scenarios, including sensitive, backdoor, and biased features and only consumes a few epochs of optimization. Thirdly, theoretical analysis in Sec. 4.3 elucidates that our proposed Ferrari achieves lower model utility loss compared to the exact feature unlearning.

The key contributions of this work are summarized as follows:

- We identify two key challenges for feature unlearning in FL. The first is how to successfully unlearn features without requiring the participation of other clients, as discussed in Sec. 3.2. The second is how to design an effective evaluation method in federated feature unlearning.

- We define the feature sensitivity and introduce this metric in federated feature unlearning in Sec. 4. By minimizing feature sensitivity, we propose an effective federated feature unlearning method, named Ferrari, which enables clients to selectively unlearn specific features from the trained global model without requiring the participation of other clients.

- We provide a theoretical proof in Theorem 1, which dictates that *Ferrari achieves better model performances than exact feature unlearning*. This analytical result is also echoed in the empirical evidence, highlighting Ferrari's effectiveness across various settings, including the unlearning of sensitive, backdoor, and biased features.

## 2 Related Work

**Machine Unlearning**  Machine Unlearning (MU), introduced by Cao et al. [23], involves selectively removing specific training data from a trained model without retraining from scratch[24, 25]. It categorizes into exact unlearning [26, 27], aiming to completely remove data influence with techniques like SISA [28] and ARCANE [29], though with computational costs, and approximate unlearning [30, 31], which reduces data impact through techniques like data manipulation (fine-tuning with mislabeled data [32–36] or introducing noise [37–39]), knowledge distillation [40–43] (training a student model), gradient ascent [44–47] (maximizing loss associated with forgotten data), and weight scrubbing [48–53] (discarding heavily influenced weights).

**Federated Unlearning**  In FL, traditional centralized MU methods are unsuitable due to inherent differences like incremental learning and limited dataset access [54]. Research on Federated Unlearning (FU) mainly focuses on client, class, and sample unlearning [6–8]. Client unlearning, pioneered by Liu et al. [55] introducing FedEraser [55], includes approaches like FRU [56], FedRecover [57], VeriFI [58], HDUS [59], KNOT [60], FedRecovery [61], Knowledge Distillation [54], and Gradient Ascent [62–64], aiming to remove specific clients or recover poisoned global models. Class unlearning, introduced by Wang et al. [65], involves frameworks like discriminative pruning and Momentum Degradation [66] (MoDE) to remove entire data classes. Sample unlearning, initiated by Liu et al. [67], targets individual sample removal within FL settings, with advancements like the QuickDrop [68] framework and FedFilter [69] enhancing efficiency and effectiveness. Recent works, such as $FedMe^2$ by Xia et al. [70], optimize both unlearning facilitation and privacy guarantees.

Existing literature on FU primarily focuses on client, class, or sample unlearning [6–8]. However, a significant gap arises when a client seeks to remove only sensitive features while remaining engaged in FL. Unfortunately, current FU approaches do not address this specific scenario, as they do not explore feature unlearning within FL settings. In contrast to prior works focusing on feature unlearning in centralized settings of MU, such as classification models [9, 10], generative models [71–74], and large language models [75–77], this study uniquely addresses feature unlearning of classification model within the FL paradigm. This distinction arises because traditional feature unlearning methods in centralized settings of MU are impractical for FL scenarios, where participation from all clients is often infeasible. In such cases, the process fails if even a single client opts out of the operation.

Therefore, to fill this critical gap, we proposed a novel federated feature unlearning framework, namely Ferrari based on the concept of Lipschitz continuity [78–80]. Our proposed Ferrari requires exclusively from the target client's dataset while still preserving the model's original performance. Lipschitz continuity, a fundamental mathematical concept that measures a function's sensitivity to changes in its input variables [81–83], is central to our feature unlearning approach. For a detailed exposition of our proposed federated feature unlearning framework utilizing Lipschitz continuity, please refer to Sec. 4. To the best of our knowledge, this is the **first work** in feature unlearning within FL settings that does not necessitate participation from all other clients, showcasing the potential to enhance privacy, practicality and efficiency.

## 3 Challenges on Feature Unlearning in FL

### 3.1 Federated Feature Unlearning

Consider a federated system comprising $K$ clients and one server, collaboratively learning a global model $f_\theta$ as:

$$\min_\theta \sum_{k=1}^{K} \sum_{i=1}^{n_k} \frac{\ell(f_\theta(x_{k,i}), y_{k,i})}{n_1 + \cdots + n_K}, \tag{1}$$

where $\ell$ is the loss, *e.g.*, the cross-entropy loss, $\mathcal{D}_k = \{(x_{k,i}, y_{k,i})\}_{i=1}^{n_k}$ is the dataset with size $n_k$ owned by client $k$. One client (*i.e.*, referred to as the unlearn client $C_u$) requests the removal of a feature $\mathcal{F}$ from the global model $\theta$ such that $\theta$ does not retain any information about $\mathcal{F}$. Specifically, we assume that the data $x \in \mathbb{R}^d$ and denote the j-th feature of $x$ by $x[j]$. The partial element of the data $x$ corresponding the feature $\mathcal{F}$ is defined as $x[\mathcal{F}]$, *i.e.,*:

$$x[\mathcal{F}] = \{x[j], j \in \mathcal{F}\} \tag{2}$$

Therefore, the unlearn client $C_u$ aims to remove $\{x_{i,u}[\mathcal{F}]\}_{i=1}^{n_u}$, called unlearned data $\mathcal{D}_u$. Denote $\mathcal{D}_r = \mathcal{D} - \mathcal{D}_u$ to be the remaining data.

### 3.2 Challenges for Feature Unlearning in FL

Unlike sample or class unlearning [6–8], evaluating the unlearning effectiveness for feature unlearning is difficult. Typically, unlearning effectiveness is assessed by comparing the unlearned model with a retrained model trained on remaining data $\mathcal{D}_r$. However, building $\mathcal{D}_r$ for the feature unlearning takes much work. For example, suppose we want to remove the mouth from a face image. In that case, one possible solution is to replace the mouth region with Gaussian noise or black block, as illustrated in Fig. 1. However, this added Gaussian noise or black block can adversely affect model training and degrade performance, *e.g.*, the degradation of model accuracy is beyond 27%.

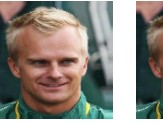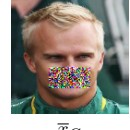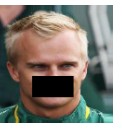

| $x$ | $\overline{x}_G$ | $\overline{x}_B$ |
| Acc 95.86% | 75.51% | 68.37% |

Figure 1: Sample data $x$ with Gaussian noise ($\overline{x}_G$) and black pixels ($\overline{x}_B$) perturbations, illustrating feature removal and performance comparison.

Another challenge is implementing feature unlearning for $C_u$ without the help of other clients. Previous work on feature unlearning [9, 10] typically requires access to the remaining data, necessitating the participation of other clients in the FL process. This requirement is impractical in the FL context, as other clients may be unwilling or unable to share data or computational resources. Therefore, finding a method to effectively unlearn features without relying on other clients is crucial to maintain the model accuracy and practicality in the FL settings.

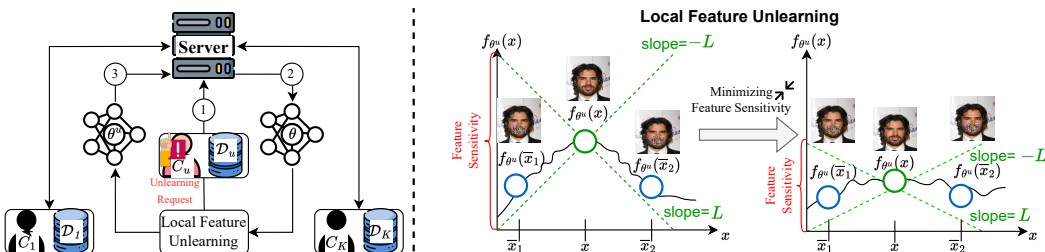

Figure 2: Overview of our proposed Ferrari framework: Initiated by the feature unlearning request from the unlearn client $C_u$, the server initializes the trained global model $\theta$ to $C_u$ for local feature unlearning. Upon completion, $C_u$ uploads the unlearned model $\theta^u$ to the server. Local feature unlearning minimizes the Lipschitz constant $L$ between the original input and its perturbed feature subset, reducing feature sensitivity yet preserving the overall model performance.

## 4 The Proposed Method

In this section, we introduce feature sensitivity (see Def. 1) in Sec. 4.1 to evaluate the effectiveness of feature unlearning. We then propose Ferrari based on this concept in Sec. 4.2). Finally, we demonstrate that Ferrari achieves a lower utility loss compared to exact feature unlearning in Sec. 4.3).

### 4.1 Feature Sensitivity

Inspired by Lipschitz Continuity [79, 80, 82], which provides an approximate method for removing information from images by perturbing the input data and observing the effect on the output, we introduce the concept of **feature sensitivity** $s$ as Def. 1. This metric measures the memorization of a model $f_\theta$ for the feature $\mathcal{F}$ by considering the local changes in the given input rather than the global change as defined in the traditional Lipschitz continuity.

**Definition 1.** *The feature sensitivity $s$ of the model $f$ with respect to the feature $\mathcal{F}$ on the data $(x, y)$ is defined as:*

$$s = \mathbb{E}_{\delta_{\mathcal{F}}} \frac{\|f(x) - f(x + \delta_{\mathcal{F}})\|_2}{\|\delta_{\mathcal{F}}\|_2}, \qquad (3)$$

*where $\delta_{\mathcal{F}}$ denote the perturbation on feature $\mathcal{F}$.*

Def. 1 characterizes the rate of change or sensitivity of the model output to perturbations in the input data. A small feature sensitivity $s$ represents the model $f$ doesn't memorize the feature $\mathcal{F}$. This definition does not require building the

---

**Algorithm 1** Federated Feature Unlearning

**Input:** Unlearn client $C_u$, Local dataset $\mathcal{D}_u$ with data size $n_u$, Unlearn feature $\{\mathcal{F}_i\}_{i=1}^N$, Global model parameters $\theta$, Gaussian noise $\sigma$, Learning rate $\eta$, Sample number $N$
**Output:** Unlearned model parameters $\theta^u$

1: ▷ *The unlearn client $C_u$ performs:*
2: **for** $(x, \mathcal{F}_i)$ in $(\mathcal{D}_u, \{\mathcal{F}_i\}_{i=1}^N)$ **do**
3:     $\theta^u = \theta$
4:     **for** $i = 1$ **to** $N$ **do**
5:         Sample $\delta_{\mathcal{F}_i}$ according to Eq. (4)
6:         Compute $L_i = \frac{\|f_{\theta^u}(x) - f_{\theta^u}(x + \delta_{\mathcal{F},i})\|_2}{\|\delta_{\mathcal{F},i}\|_2}$
7:     **end for**
8:     $L = \frac{1}{N} \sum_{i=1}^N L_i$
9:     $\theta^u \leftarrow \theta^u - \eta \cdot \nabla_{\theta^u}(L)$
10: **end for**
11: Upload $\theta^u$ to the server
12: ▷ *The server performs:*
13: Replace the global model $\theta$ with the $\theta^u$
14: **return** $\theta^u$

---

remaining data, as it considers the expectation over the perturbation $\delta_{\mathcal{F}}$. Specifically, it represents the average output change rate over any magnitude of the perturbation. Furthermore, we will provide the relationship between Def. 1 and exact feature unlearning in Sec. 4.3.

**Remark 1.** *The perturbation $\delta_{\mathcal{F}}$ can be chosen from various distributions, such as the Gaussian distribution, the uniform distribution, and so on.*

### 4.2 Ferrari

As discussed the feature sensitivity $s$ in Sec. 4.1, the core idea of the proposed method Ferrari is to achieve the feature unlearning by minimizing the feature sensitivity. More specifically, it controls the change in the model's output relative to changes in the input within the feature region, *i.e.,* the slope, to prevent the model from memorizing the feature as illustrated in Fig. 2.

One unlearning client $C_u$ requests to unlearning the feature $\mathscr{F}$. The proposed Ferrari aims to unlearn the global model $\theta$ to $\theta^u$. The proposed method can be divided into three steps (see details in Alg. 1). In order to compute the feature sensitivity, the perturbation $\delta_{\mathscr{F}}$ in terms of the feature $\mathscr{F}$ is **firstly** computed as the following (take the Gaussian distribution as an example):

$$\delta_{\mathscr{F}}[j] = \begin{cases} \sim N(0, \sigma^2) & j \in \mathscr{F} \\ 0 & \text{Otherwise} \end{cases} \tag{4}$$

**Secondly,** we leverage a finite sample Monte Carlo approximation to the maximization as Def. 1 as:

$$\mathbb{E}_{\delta_{\mathscr{F}}} \frac{\|f_\theta(x) - f_\theta(x + \delta_{\mathscr{F}})\|_2}{\|\delta_{\mathscr{F}}\|_2} \sim \frac{1}{N} \sum_{i=1}^{N} \frac{\|f_\theta(x) - f_\theta(x + \delta_{\mathscr{F},i})\|_2}{\|\delta_{\mathscr{F},i}\|_2}, \tag{5}$$

where $\delta_{\mathscr{F},i}$ is $i_{th}$ sampling as Eq. (4).

**Finally**, for the unlearning client $C_u$ who aims to remove the feature $\mathscr{F}$ from his/her data $\mathscr{D}_u$, the unlearned model $\theta^u$ is obtained as the following:

$$\theta^u = \arg\min_\theta \mathbb{E}_{(x,y) \in \mathscr{D}_u} \frac{1}{N} \sum_{i=1}^{N} \frac{\|f_\theta(x) - f_\theta(x + \delta_{\mathscr{F},i})\|_2}{\|\delta_{\mathscr{F},i}\|_2}, \tag{6}$$

where Eq. (6) is computed over the dataset $\mathscr{D}_u$. Noted that the proposed Ferrari based on Def. 1 doesn't need the participation of other clients.

**Remark 2.** *When the unlearning happens during the federated training, the unlearning clients would also optimize the training loss and feature sensitivity simultaneously, i.e.,, $\mathbb{E}_{(x,y) \in \mathscr{D}}\big(\ell(f_\theta(x), y) + \lambda \mathbb{E}_{\delta_{\mathscr{F}}} \frac{\|f_\theta(x) - f_\theta(x + \delta_{\mathscr{F}})\|_2}{\|\delta_{\mathscr{F}}\|_2}\big)$, where $\lambda$ is a coefficient.*

### 4.3 Theoretical Analysis of the Utility loss for Ferrari

As illustrated in Sec. 3.2, retraining the model without the feature may affect the model accuracy seriously. Suppose the feature is successfully removed when the norm of perturbation is larger than $C$. We firstly define the utility loss $\ell_1$ with unlearning feature directly, *i.e.,*, **the exact feature unlearning**:

$$\ell_1 = \min_{\|\delta_{\mathscr{F}}\| \geq C} \mathbb{E}_{(x,y) \in \mathscr{D}} \min_\theta \ell\big(f_\theta(x + \delta_{\mathscr{F}}), y\big) \tag{7}$$

And we define the maximum utility loss with the norm perturbation lower than $C$ as:

$$\ell_2 = \max_{\|\delta_{\mathscr{F}}\| \leq C} \mathbb{E}_{(x,y) \in \mathscr{D}} \min_\theta \ell\big(f_\theta(x + \delta_{\mathscr{F}}), y\big) \tag{8}$$

**Assumption 1.** *Assume $\ell_2 \leq \ell_1$*

Assumption 1 elucidates that the utility loss associated with a perturbation norm lower than $C$ is smaller than the utility loss when the perturbation norm is greater than $C$. This assumption is logical, as larger perturbations would naturally lead to a greater utility loss.

**Assumption 2.** *Suppose the federated model achieves zero training loss.*

We have the following theorem to elucidate the relation between feature sensitivity removing via Alg. 1 and exact unlearning (see proof in Appendix A.1, including the extension for the non-zero training loss assumption).

**Theorem 1.** *If Assumptions 1 and 2 hold, the utility loss of unlearned model obtained using Alg. 1 is lower than the utility loss with exact feature unlearning, i.e.,,*

$$\ell_u \leq \ell_1, \tag{9}$$

*where $\ell_u = \mathbb{E}_{(x,y) \in \mathscr{D}} \ell(f_{\theta^u}(x), y)$*

Theorem 1 showcases that the proposed method Ferrari, results in a utility loss ($\ell_u$) that is lower than the utility loss incurred when the feature is removed, and the model is retrained, *i.e.,* the process of exact feature unlearning.

**Remark 3.** *To further evaluate the effectiveness of feature unlearning based on feature sensitivity, we employ model inversion attacks [11, 12] to determine if the feature can be reconstructed and employ attention maps to assess if the model still focuses on the unlearned feature, as described in Sec. 5.3.1.*

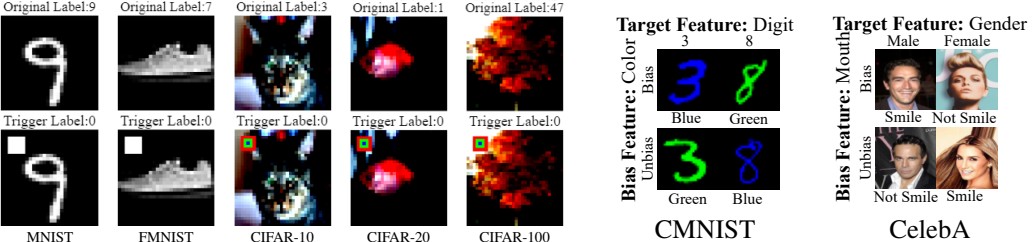

Figure 3: Pixel-pattern backdoor feature.    Figure 4: Biased datasets distribution.

# 5 Experimental Results

This section presents the empirical analysis of the proposed Ferrari framework in terms of effectiveness, utility, and time efficiency in sensitive, backdoor and biased feature unlearning scenarios.

## 5.1 Experimental Setup

**Unlearning Scenarios** *Sensitive Feature Unlearning*: We simulate the removal of sensitive features from the $\mathcal{D}_u$ to fulfill the request of $C_u$ due to privacy concern. Specifically, we remove 'mouth' from CelebA [84], 'marital status' from Adult [85], and 'pregnancies number' from Diabetes [86]. Therefore, our proposed Ferrari aims to remove the influence of these requested features.

*Backdoor Feature Unlearning*: We simulate a pixel-pattern backdoor attack by $C_u$ based on BadNets [18] within a FL framework [15–17]. $C_u$ injects a pixel-pattern backdoor feature and trigger label into its $\mathcal{D}_u$ during training, as shown in Fig. 3. Consequently, our proposed Ferrari aims to remove the influence of these backdoor features and restore the model's original performance.

*Biased Feature Unlearning*: We simulate the bias dataset $\mathcal{D}_u$ of the $C_u$ and the unbias dataset $\mathcal{D}_r$ with a bias ratio of 0.8, as shown in Fig. 4. This results in a global model biased towards the biased dataset [87, 88] due to unintended feature memorization [22]. In CMNIST [89], the model focuses on color patterns instead of digits, and in CelebA [84], it learns mouth features instead of facial features for gender classification. Therefore, our proposed Ferrari aims to mitigate these bias-inducing features and restore model performance.

**Hyperparameters & Datasets & Model** We simulate HFL with $K = 10$ clients under an IID setting, each holding $10\%$ of the datasets, except for the biased feature unlearning experiment with a bias ratio of 0.8. For federated feature unlearning experiments, we set hyperparameters: learning rate $\eta = 0.0001$, sample size $N = 20$, and random Gaussian noise with standard deviation ranging from $0.05 \leq \sigma \leq 1.0$ (see Sec. 5.5) across iterations of $N$. Experiments are repeated over five random trials, and results are reported as mean and standard deviation. We employ ResNet18 [90] on image datasets: MNIST [89], Colored-MNIST (CMNIST) [89], Fashion-MNIST [91], CIFAR-10, CIFAR-20, CIFAR-100 [92] and ImageNet [93]. For tabular datasets, such as Adult Census Income (Adult) [85] and Diabetes [86], we used a fully-connected neural network linear model. Additionally, we utilize the transformer-based BERT model [94] for the text dataset, specifically the IMDB movie reviews dataset [95]. We conduct experiments on a single NVIDIA A100 GPU. Further details are in Appendix A.2.

**Evaluation Metrics** We assess effectiveness by measuring feature sensitivity (see Section 4.1) and conducting a model inversion attack (MIA) [11–14] to determine the attack success rate (ASR). The goal is to achieve low feature sensitivity and ASR, indicating successful unlearning sensitive features. Backdoor and biased feature unlearning are evaluated by comparing accuracy on the retain dataset $\mathcal{D}_r$ ($Acc_r$) and the unlearn client dataset $\mathcal{D}_u$($Acc_u$). Low $Acc_u$ indicates high effectiveness for backdoor unlearning, while similar accuracy ($Acc_r \approx Acc_u$) reflects fairness and effectiveness in biased feature unlearning. Qualitatively, effectiveness is assessed using MIA-reconstructed images (sensitive) and GradCAM [96] attention maps (backdoor and biased). The utility is measured by test dataset $\mathcal{D}_t$ accuracy ($Acc_t$), with higher values indicating stronger utility. Time efficiency is evaluated by comparing the runtime of each baseline.

| Scenarios | Datasets | Unlearn Feature | Accuracy(%) | | | | | |
|---|---|---|---|---|---|---|---|---|
| | | | **Baseline** | **Retrain** | **Fine-tune** | **FedCDP[65]** | **FedRecovery[61]** | **Ferrari (Ours)** |
| **Sensitive** | **CelebA** | Mouth | 94.87 ±1.38 | 79.46 ±2.32 | 62.79 ±1.62 | 34.03 ±4.20 | 29.78 ±6.69 | **92.26 ±1.73** |
| | **Adult** | Marriage | 82.45 ±2.59 | 65.27 ±0.58 | 61.02 ±1.05 | 30.19 ±1.62 | 27.89 ±3.71 | **81.02 ±0.58** |
| | **Diabetes** | Pregnancies | 82.11 ±0.49 | 64.19 ±0.72 | 59.57 ±0.68 | 36.71 ±4.56 | 17.56 ±2.32 | **79.53 ±0.79** |
| | **IMDB** | Names | 91.39 ±1.57 | 83.27 ±2.05 | 72.15 ±1.92 | 48.36 ±2.79 | 37.93 ±2.84 | **89.15 ±1.32** |
| **Backdoor** | **MNIST** | Backdoor Pixel Pattern | 94.75 ±4.88 | 96.23 ±0.16 | **96.85 ±0.91** | 65.31 ±4.39 | 40.52 ±7.38 | 95.83 ±1.14 |
| | **FMNIST** | | 90.68 ±2.19 | 92.98 ±0.75 | **93.52 ±1.63** | 67.62 ±0.81 | 42.24 ±4.45 | 92.61 ±1.57 |
| | **CIFAR-10** | | 87.55 ±3.71 | 90.92 ±1.83 | **91.23 ±0.44** | 53.98 ±2.17 | 27.16 ±9.68 | 89.52 ±2.18 |
| | **CIFAR-20** | | 74.47 ±2.38 | 81.61 ±1.75 | **82.52 ±0.69** | 54.76 ±0.98 | 23.02 ±3.11 | 78.34 ±2.35 |
| | **CIFAR-100** | | 54.13 ±7.62 | 73.12 ±1.54 | **73.59 ±1.66** | 34.30 ±0.42 | 15.21 ±5.83 | 69.30 ±2.27 |
| | **ImageNet** | | 52.86 ±4.14 | 67.18 ±2.07 | **67.52 ±1.69** | 31.17 ±3.96 | 12.75 ±5.27 | 65.36 ±1.84 |
| **Biased** | **CMNIST** | Color | 81.72 ±3.41 | 98.49 ±1.46 | 82.54 ±0.78 | 27.56 ±1.71 | 25.05 ±5.09 | **83.85 ±1.63** |
| | **CelebA** | Mouth | 87.35 ±4.07 | 95.87 ±1.52 | 88.93 ±2.65 | 16.98 ±0.23 | 20.19 ±7.21 | **94.62 ±2.49** |

Table 1: The accuracy of $\mathcal{D}_t$ for each unlearning method across different unlearning scenarios.

| Scenario | Datasets | Unlearn Feature | Feature Sensitivity | | | | | |
|---|---|---|---|---|---|---|---|---|
| | | | **Baseline** | **Retrain** | **Fine-tune** | **FedCDP [65]** | **FedRecovery [61]** | **Ferrari (Ours)** |
| **Sensitive** | **CelebA** | Mouth | $0.96 \pm 1.41 \times 10^{-2}$ | $0.07 \pm 8.06 \times 10^{-4}$ | $0.79 \pm 2.05 \times 10^{-2}$ | $0.93 \pm 2.87 \times 10^{-2}$ | $0.91 \pm 3.41 \times 10^{-2}$ | $\mathbf{0.09 \pm 3.04 \times 10^{-4}}$ |
| | **Adult** | Marriage | $1.31 \pm 1.53 \times 10^{-2}$ | $0.02 \pm 6.47 \times 10^{-4}$ | $0.94 \pm 6.81 \times 10^{-2}$ | $1.07 \pm 7.43 \times 10^{-2}$ | $1.14 \pm 2.57 \times 10^{-2}$ | $\mathbf{0.05 \pm 1.72 \times 10^{-4}}$ |
| | **Diabetes** | Pregnancies | $1.52 \pm 0.91 \times 10^{-2}$ | $0.05 \pm 5.07 \times 10^{-4}$ | $0.96 \pm 1.28 \times 10^{-2}$ | $1.23 \pm 3.82 \times 10^{-2}$ | $0.83 \pm 5.08 \times 10^{-2}$ | $\mathbf{0.07 \pm 1.07 \times 10^{-4}}$ |
| | **IMDB** | Names | $0.85 \pm 1.07 \times 10^{-2}$ | $0.07 \pm 5.38 \times 10^{-4}$ | $0.74 \pm 3.81 \times 10^{-2}$ | $0.81 \pm 3.27 \times 10^{-2}$ | $0.78 \pm 2.41 \times 10^{-2}$ | $\mathbf{0.08 \pm 1.32 \times 10^{-4}}$ |

Table 2: Feature sensitivity for each unlearning method across sensitive feature unlearning scenario.

**Baselines**    We compare our proposed Ferrari against the models of Baseline, Retrain, Fine-tune, FedCDP [65] and FedRecovery [61]. Additional details are provided in Appendix A.2.

## 5.2    Utility Guarantee

To evaluate the utility of Ferrari, we measure $Acc_t$ on $\mathcal{D}_t$, where a higher $Acc_t$ indicates greater utility (Tab. 1). Although the Fine-tune method shows high $Acc_t$ in the backdoor feature unlearning scenario with a clean dataset, its unlearning effectiveness is very low (see Sec. 5.3.2). This problem worsens with FedCDP [65] and FedRecovery [61], which suffer significant $Acc_t$ declines, reducing model utility and making them unsuitable for feature unlearning. In contrast, Ferrari achieves the highest model utility in sensitive and biased feature unlearning scenarios, with the highest $Acc_t$ among baselines, minimal deterioration, and the greatest unlearning effectiveness across all scenarios.

## 5.3    Effectiveness Guarantee

In this subsection, we analyze the unlearning effectiveness of Ferrari against baselines in sensitive, backdoor, and biased feature unlearning scenarios.

### 5.3.1    Sensitive Feature Unlearning

To evaluate Ferrari's effectiveness in unlearning sensitive features, we measured feature sensitivity (see Sec. 4.1) and conducted a model inversion attack (MIA) [11–14].

**Feature Sensitivity**    Tab. 2 shows the sensitivity of the unlearn feature. The baseline model had high sensitivity to this feature. Similar results were observed for the Fine-tune, FedCDP [65], and FedRecovery models [61], with sensitivities greater than 0.8, indicating ineffective unlearning. In contrast, our proposed Ferrari model exhibits low sensitivity, similar to the Retrain model, indicating successful unlearning of the sensitive feature.

**ASR of MIA**    Tab. 3 shows the ASR results. The Baseline model achieved an ASR exceeding 80%, indicating substantial exposure of sensitive features. Similar observations were made for the Fine-tune, FedCDP [65], and FedRecovery [61] models, with ASR surpassing 70% exhibiting ineffective feature unlearning. Conversely, Ferrari achieved low ASR, suggesting successful feature unlearning with minimal unlearned feature exposure after using Ferrari via MIA.

**MIA Reconstruction**    Fig. 5 shows MIA-reconstructed images. The Baseline model achieved complete reconstruction, whereas both Retrain and Ferrari models failed to reconstruct the mouth feature accurately. This underscores Ferrari's effectiveness in unlearning and preserving privacy by preventing precise reconstruction of unlearned features via MIA.

| Scenario | Datasets | Unlearn Feature | Attack Success Rate(ASR) (%) | | | | | |
|---|---|---|---|---|---|---|---|---|
| | | | Baseline | Retrain | Fine-tune | FedCDP [65] | FedRecovery [61] | Ferrari (Ours) |
| Sensitive | CelebA | Mouth | 84.36 ±3.22 | 47.52 ±1.04 | 77.43 ±10.98 | 75.36 ±9.31 | 71.52 ±6.07 | **51.28 ±2.41** |
| | Adult | Marriage | 87.54 ±13.89 | 49.28 ±2.13 | 83.45 ±8.44 | 72.83 ±5.18 | 80.39 ±10.68 | **49.58 ±1.38** |
| | Diabetes | Pregnancies | 92.31 ±7.55 | 38.89 ±2.52 | 88.46 ±5.01 | 81.91 ±8.17 | 78.27 ±2.47 | **42.61 ±1.81** |
| | IMDB | Names | 90.28 ±2.49 | 40.29 ±1.59 | 86.74 ±3.81 | 83.67 ±4.59 | 80.95 ±3.51 | **43.75 ±1.86** |

Table 3: The ASR of MIA for each unlearning method across sensitive feature unlearning scenario.

| Target | Baseline | Retrain | Ferrari (Ours) | Target | Baseline | Retrain | Ferrari (Ours) |
|---|---|---|---|---|---|---|---|

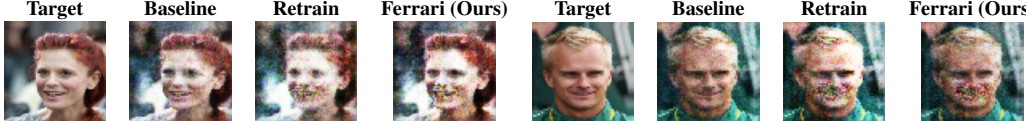

Figure 5: MIA reconstruction on CelebA (unlearned mouth)

### 5.3.2 Backdoor Feature Unlearning

**Accuracy** $\mathcal{D}_r$ and $\mathcal{D}_u$ represent the clean and backdoor datasets, respectively. Successful unlearning is shown by low $Acc_u$ and high $Acc_r$, indicating effective unlearning and preserved model utility. As shown in Tab. 4, the Fine-tune method has higher $Acc_r$ and utility than the Retrain method but lower unlearning effectiveness due to high $Acc_u$. FedCDP [65] and FedRecovery [61] show low utility and unlearning effectiveness with low $Acc_r$ and $Acc_u$, rendering them unsuitable for backdoor feature unlearning. In contrast, Ferrari demonstrates the highest utility and unlearning effectiveness.

**Attention Map** Fig. 6a illustrates attention maps analyzing backdoor feature unlearning. Initially, the Baseline model focuses on the $5 \times 5$ square at the top-left corner, indicating a significant influence on output prediction by the pixel-pattern backdoor feature. In contrast, Ferrari unlearned models shift the attention towards recognizable objects like digits and cars, similar to the Retrain model. This shift suggests a reduced sensitivity to the backdoor feature, indicating a successful unlearning. See Appendix A.3.1 for supplementary results.

### 5.3.3 Biased Feature Unlearning

**Accuracy** $\mathcal{D}_r$ and $\mathcal{D}_u$ represent the unbias and bias datasets, respectively. Successful unlearning results in similar accuracies across both datasets ($Acc_r \approx Acc_u$), ensuring fairness while maintaining high $Acc_r$ and $Acc_u$ for utility. Tab. 4 shows that the Fine-tune method fails to unlearn bias, as $Acc_u$ remains higher than $Acc_r$, despite slightly higher $Acc_r$ compared to Retrain. FedCDP [65] and FedRecovery [61] exhibit catastrophic forgetting, with low $Acc_r$ and $Acc_u$, making them unsuitable for biased feature unlearning. In contrast, Ferrari demonstrates effective unlearning with similar $Acc_r$ and $Acc_u$, and maintains high overall accuracy, indicating a successful biased feature unlearning.

**Attention Map** Fig. 6b shows attention maps analyzing biased feature unlearning. The Baseline model predominantly focuses on the biased feature region (mouth) in both bias and unbias datasets, suggesting its significant impact on output prediction. However, Ferrari unlearned models redistribute attention across various facial regions in both datasets, similar to the Retrain model. This shift indicates reduced sensitivity to the biased feature, demonstrating successful unlearning. See Appendix A.3.2 for supplementary results.

### 5.4 Computational Complexity

In Fig. 7, we evaluate the runtime performance and FLOPs metrics of each unlearning method to demonstrate the computational complexity. The Retrain method is expected to have the slowest runtime and highest FLOPs, while Fine-tune is fast but still slower than other methods.

Both FedCDP [65] and FedRecovery [61] demonstrate faster runtimes and lower FLOPs than the Fine-tune method, but they are still more computationally expensive than Ferrari. This is primarily due to the need to access training datasets from all clients and the computational expense of gradient residual calculations [61].

| Scenarios | Datasets | Unlearn Feature | Accuracy (%) | | | | | |
|---|---|---|---|---|---|---|---|---|
| | | | Baseline | Retrain | Fine-tune | FedCDP[65] | FedRecovery[61] | Ferrari(Ours) |
| Backdoor | MNIST | Backdoor pixel-pattern | $\mathcal{D}_r$ 95.65 ±1.39 | 97.19 ±2.49 | **96.16 ±0.37** | 65.82 ±6.85 | 40.81 ±4.31 | 95.93 ±0.45 |
| | | | $\mathcal{D}_u$ 97.43 ±3.69 | 0.00 ±0.00 | 69.37 ±0.83 | 72.64 ±0.24 | 53.72 ±3.14 | **0.11 ±0.01** |
| | FMNIST | | $\mathcal{D}_r$ 91.07 ±0.54 | 93.85 ±1.08 | **94.36 ±1.98** | 68.46 ±3.39 | 42.93 ±2.50 | 92.83 ±0.61 |
| | | | $\mathcal{D}_u$ 94.51 ±6.29 | 0.00 ±0.00 | 43.91 ±0.28 | 72.19 ±0.49 | 48.15 ±4.37 | **0.90 ±0.03** |
| | CIFAR-10 | | $\mathcal{D}_r$ 87.63 ±1.16 | 91.12 ±1.60 | **92.02 ±3.15** | 54.91 ±6.91 | 27.49 ±4.96 | 89.91 ±0.95 |
| | | | $\mathcal{D}_u$ 95.05 ±2.30 | 0.00 ±0.00 | 88.44 ±0.92 | 62.75 ±5.07 | 49.26 ±2.23 | **0.29 ±0.04** |
| | CIFAR-20 | | $\mathcal{D}_r$ 75.06 ±6.41 | 81.91 ±4.68 | **82.67 ±1.32** | 55.67 ±6.35 | 23.76 ±2.17 | 78.29 ±3.12 |
| | | | $\mathcal{D}_u$ 94.21 ±4.11 | 0.00 ±0.00 | 86.53 ±1.47 | 50.17 ±9.11 | 50.38 ±4.25 | **0.78 ±0.08** |
| | CIFAR-100 | | $\mathcal{D}_r$ 54.14 ±3.96 | 73.54 ±5.70 | **73.66 ±6.57** | 34.62 ±2.24 | 15.62 ±7.78 | 69.57 ±3.81 |
| | | | $\mathcal{D}_u$ 88.98 ±6.63 | 0.00 ±0.00 | 65.38 ±4.76 | 57.29 ±3.62 | 46.17 ±9.25 | **0.15 ±0.01** |
| | ImageNet | | $\mathcal{D}_r$ 52.35 ±2.25 | 67.05 ±1.29 | **67.34 ±2.73** | 29.74 ±4.72 | 13.46 ±6.53 | 65.74 ±1.32 |
| | | | $\mathcal{D}_u$ 83.16 ±3.74 | 0.00 ±0.00 | 71.48 ±3.69 | 62.39 ±3.05 | 54.92 ±5.59 | **0.09 ±0.02** |
| Biased | CMNIST | Color | $\mathcal{D}_r$ 64.94 ±7.88 | 98.76 ±3.65 | 67.15 ±2.60 | 25.85 ±1.58 | 23.92 ±1.08 | **84.31 ±2.63** |
| | | | $\mathcal{D}_u$ 98.88 ±4.90 | 98.44 ±1.90 | 97.95 ±1.13 | 30.17 ±4.69 | 27.64 ±9.37 | **84.62 ±3.59** |
| | CelebA | Mouth | $\mathcal{D}_r$ 79.46 ±2.09 | 96.47 ±6.15 | 84.45 ±1.48 | 14.29 ±0.81 | 16.34 ±3.43 | **94.18 ±3.08** |
| | | | $\mathcal{D}_u$ 96.38 ±3.87 | 96.11 ±2.17 | 94.23 ±0.66 | 21.58 ±3.48 | 25.72 ±8.02 | **94.79 ±1.48** |

Table 4: The accuracy of $\mathcal{D}_r$ and $\mathcal{D}_u$ for each unlearning method across different unlearning scenarios.

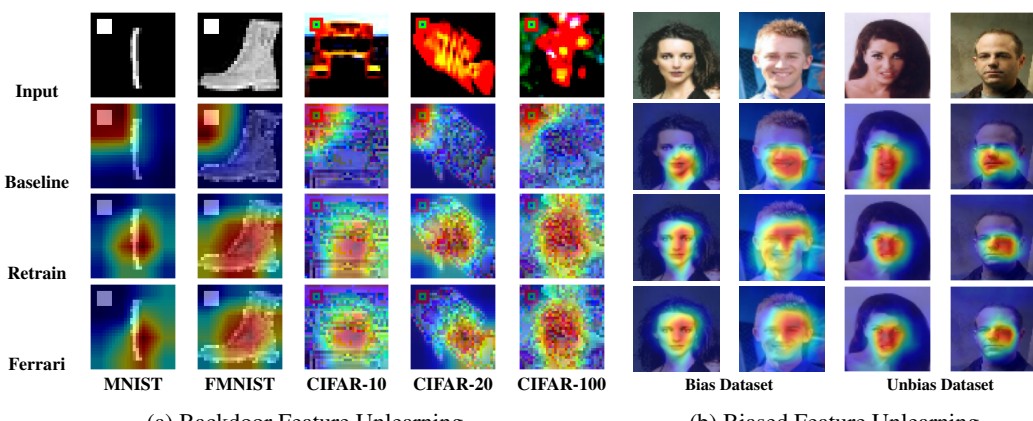

(a) Backdoor Feature Unlearning  (b) Biased Feature Unlearning

Figure 6: The attention map of each unlearning method across different unlearning scenarios.

In contrast, Ferrari has the lowest computational complexity, with the fastest runtime and lowest FLOPs. It only requires access to the local dataset of the unlearn client and achieves feature unlearning by minimizing feature sensitivity within a single epoch.

## 5.5 Ablation Study and Hyper-parameter Analysis

We conduct an ablation study to analyze how Non-Lipschitz affects the effectiveness of our proposed Ferrari and hyper-parameter analysis of Gaussian noise level ($\sigma$) and number of $\mathcal{D}_u$ in Fig. 8.

**Non-Lipschitz**   We evaluate the unlearning performance by removing the denominator in Eq. 6, calling this the Non-Lipschitz method, as shown in Fig. 8a. The results indicate catastrophic forgetting: $\mathcal{D}_r$ accuracy drops below 10%, and the unlearned model misclassifies all inputs into a single random class, rendering it useless. This stems from the unbounded loss function in the non-Lipschitz method, unlike the bounded Lipschitz constant in Eq. 6, which provides a theoretical guarantee (see Sec. 4.3). Refer to Appendix A.4 for a detailed analysis of Lipschitz and Non-Lipschitz loss functions.

**Gaussian Noise**   The effectiveness of Ferrari is significantly influenced by injected Gaussian noise. Fig. 8b shows the accuracy of $\mathcal{D}_r$ and $\mathcal{D}_u$ across different $\sigma$ levels. In the $0.05 \leq \sigma \leq 1.0$ range, $\mathcal{D}_r$ accuracy stays high and $\mathcal{D}_u$ accuracy remains low, indicating a balance. Thus, we implement $\sigma$ values between 0.05 and 1.0 for a balanced accuracy across $\mathcal{D}_r$ and $\mathcal{D}_u$.

**Number of Unlearn Dataset**   Our analysis illustrated in Fig. 8c, demonstrates that Ferrari remains effective with partial $\mathcal{D}_u$ from $C_u$ for feature unlearning (*i.e.,* data lost). Using 70% of $\mathcal{D}_u$ yields comparable accuracy to using the full (*i.e.,* 100%) dataset, highlighting the method's flexibility even with partial data.

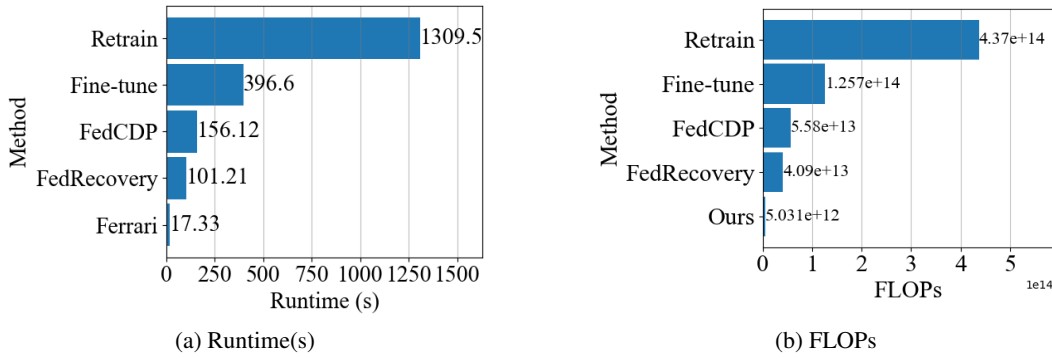

(a) Runtime(s)              (b) FLOPs

Figure 7: Computational complexity analysis comparing the runtime(s) and FLOPs for each unlearning method.

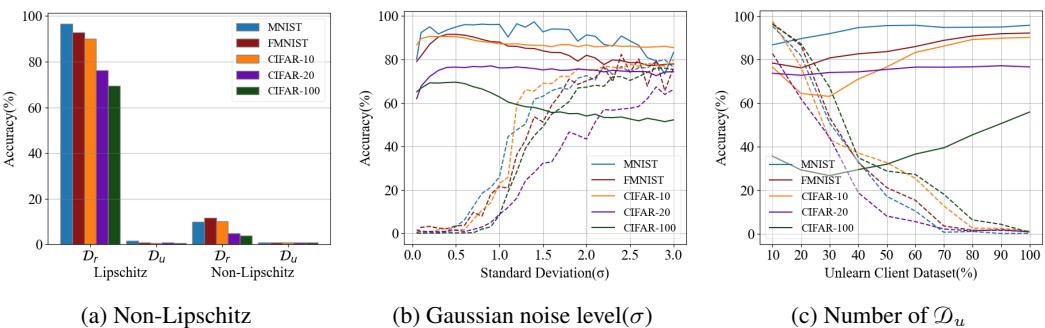

(a) Non-Lipschitz   (b) Gaussian noise level($\sigma$)   (c) Number of $\mathscr{D}_u$

Figure 8: Ablation and hyper-parameter analysis on Ferrari backdoor feature unlearning. Solid line: $\mathscr{D}_r$; dashed line: $\mathscr{D}_u$.

## 6 Conclusion

This paper introduces Ferrari, a federated feature unlearning framework designed to efficiently remove sensitive, backdoor, and biased features without extensive retraining. Leveraging Lipschitz continuity, Ferrari reduces model sensitivity to specific features, ensuring robust and fair models. Uniquely, it requires participation only from the client requesting unlearning, preserving privacy and practicality in FL environments. Experimental results and theoretical analysis demonstrate Ferrari's effectiveness across various data domains, addressing the crucial need for feature-level unlearning in federated learning. This method can serve as a technical solution to meet regulatory requirements for data deletion while maintaining model performance, offering significant value to clients by securing their "right to be forgotten" and preventing potential privacy leakage.

### 6.1 Limitation and Future Work

The proposed federated feature unlearning method works effectively using only the unlearning client's local data, making it well-suited for real-world scenarios. However, for optimal results, access to the full dataset is required. As demonstrated in Section 5.5, using 70% of the data yields comparable performance, but significant data reduction diminishes effectiveness. Future research should focus on developing methods that require only a small portion of the client's data and expanding the approach beyond classification models to include for example, generative models. Additionally, enhancements such as advanced perturbation techniques and integration with privacy-preserving methods should be explored.

## Acknowledgement

This research is supported by the Fundamental Research Grant Scheme (FRGS/1/2024/ICT02/UM/01/1), awarded by the Ministry of Higher Education, Malaysia.

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

# A   Appendix

## A.1   Proof of Theorem 1

As illustrated in Sec. 3.2, it is hard to build the unlearned data $x^u$ for the feature unlearning since adding the perturbation may influence the model accuracy seriously. Suppose the feature is successfully removed when the norm of perturbation is larger than $C$. We define the utility loss $\ell_1$ with unlearning feature successfully:

$$\ell_1 = \min_{\|\delta_{\mathcal{G}}\| \geq C} \mathbb{E}_{(x,y) \in \mathcal{D}} \min_{\theta} \ell\big(f_\theta(x + \delta_{\mathcal{G}}), y\big) \tag{10}$$

And we define the maximum utility loss with the norm perturbation less than $C$ as:

$$\ell_2 = \max_{\|\delta_{\mathcal{G}}\| \leq C} \mathbb{E}_{(x,y) \in \mathcal{D}} \min_{\theta} \ell\big(f_\theta(x + \delta_{\mathcal{G}}), y\big) \tag{11}$$

**Assumption 3.** *Assume $\ell_2 \leq \ell_1$*

Assumption 3 elucidates that the utility loss associated with a perturbation norm less than $C$ is smaller than the utility loss when the perturbation norm is greater than $C$. This assumption is logical, as larger perturbations would naturally lead to greater utility loss.

**Assumption 4.** *Suppose the federated model achieves zero training loss.*

We have the following theorem to elucidate the relation between feature sensitivity removing via Alg. 1 and exact unlearning (see proof in Appendix).

**Theorem 2.** *If Assumption 3 and 4 hold, the utility loss of unlearned model obtained by Alg. 1 is less than the utility loss with unlearning successfully, i.e.,,*

$$\ell_u \leq \ell_1, \tag{12}$$

*where $\ell_u = \mathbb{E}_{(x,y) \in \mathcal{D}}\big(\ell(f_{\theta^u}(x), y)$*

*Proof.* When the unlearning happens during the federated training, the unlearning clients would also optimize the training loss and feature sensitivity simultaneously. Specifically, the optimization process could be written as:

$$\theta_u = \arg\min_{\theta} \mathbb{E}_{(x,y) \in \mathcal{D}}\big(\ell(f_\theta(x), y) + \lambda \mathbb{E}_{\lambda \leq \|\delta_{\mathcal{G}}\| \leq C} \frac{\|f_\theta(x) - f_\theta(x + \delta_{\mathcal{G}})\|_2}{\|\delta_{\mathcal{G}}\|_2}\big),$$

where $\lambda \leq C$ is one coefficient. Without loss of generality, we assume the $\ell(f_\theta(x), y) = \|f_\theta(x) - y\|$. Denote

$$\Theta^* = \arg\min_{\theta} \mathbb{E}_{(x,y) \in \mathcal{D}} \ell(f_\theta(x), y).$$

If Assumption 4 holds, then $f_{\theta^*}(x) = y$ for any $\theta^* \in \Theta^*$. Therefore, for any $\lambda \leq \|\delta_{\mathcal{G}}\| \leq C$ such that

$$
\begin{aligned}
&\mathbb{E}_{(x,y) \in \mathcal{D}}\big(\ell(f_{\theta^*}(x), y) + \lambda \mathbb{E}_{\lambda \leq \|\delta_{\mathcal{G}}\| \leq C} \frac{\|f_\theta(x) - f_{\theta^*}(x + \delta_{\mathcal{G}})\|_2}{\|\delta_{\mathcal{G}}\|_2}\big) \\
&= \lambda \mathbb{E}_{(x,y) \in \mathcal{D}} \mathbb{E}_{\lambda \leq \|\delta_{\mathcal{G}}\| \leq C} \frac{\|y - f_{\theta^*}(x + \delta_{\mathcal{G}})\|_2}{\|\delta_{\mathcal{G}}\|_2} \\
&\leq \mathbb{E}_{(x,y) \in \mathcal{D}} \mathbb{E}_{\lambda \leq \|\delta_{\mathcal{G}}\| \leq C} \|y - f_{\theta^*}(x + \delta_{\mathcal{G}})\|_2.
\end{aligned}
\tag{13}
$$

Therefore, we further obtain:

$$
\begin{aligned}
\ell_u &= \min_{\theta \in \mathbb{R}^d} \mathbb{E}_{(x,y)\in\mathscr{D}} \Big( \ell(f_\theta(x), y) + \lambda \mathbb{E}_{\lambda \leq \|\delta_\mathscr{F}\| \leq C} \frac{\|f_\theta(x) - f_\theta(x+\delta_\mathscr{F})\|_2}{\|\delta_\mathscr{F}\|_2} \Big) \\
&\leq \min_{\theta \in \Theta^*} \mathbb{E}_{(x,y)\in\mathscr{D}} \Big( \ell(f_\theta(x), y) + \lambda \mathbb{E}_{\lambda \leq \|\delta_\mathscr{F}\| \leq C} \frac{\|f_\theta(x) - f_\theta(x+\delta_\mathscr{F})\|_2}{\|\delta_\mathscr{F}\|_2} \Big) \\
&\leq \min_{\theta \in \Theta^*} \mathbb{E}_{(x,y)\in\mathscr{D}} \mathbb{E}_{\lambda \leq \|\delta_\mathscr{F}\| \leq C} \|y - f_{\theta^*}(x+\delta_\mathscr{F})\|_2 \\
&\leq \mathbb{E}_{(x,y)\in\mathscr{D}} \mathbb{E}_{\lambda \leq \|\delta_\mathscr{F}\| \leq C} \min_{\theta \in \Theta^*} \|y - f_{\theta^*}(x+\delta_\mathscr{F})\|_2 \\
&= \mathbb{E}_{\lambda \leq \|\delta_\mathscr{F}\| \leq C} \mathbb{E}_{(x,y)\in\mathscr{D}} \min_{\theta \in \Theta^*} \|y - f_{\theta^*}(x+\delta_\mathscr{F})\|_2 \\
&\leq \max_{\lambda \leq \|\delta_\mathscr{F}\| \leq C} \mathbb{E}_{(x,y)\in\mathscr{D}} \min_{\theta \in \mathbb{R}^d} \|y - f_{\theta^*}(x+\delta_\mathscr{F})\|_2 \\
&\leq \max_{\|\delta_\mathscr{F}\| \leq C} \mathbb{E}_{(x,y)\in\mathscr{D}} \min_{\theta \in \mathbb{R}^d} \|y - f_{\theta^*}(x+\delta_\mathscr{F})\|_2 \\
&= \ell_2,
\end{aligned}
\tag{14}
$$

According to Assumption 3, we have $\ell_u \leq \ell_1$

$\square$

## A.2 Experimental Setup

**Datasets** *MNIST* [89]: Both the *MNIST* [89] and *Fashion-MNIST(FMNIST)* [91] datasets contain images of handwritten digits and attire, respectively. Each dataset comprises 60,000 training examples and 10,000 test examples. In both datasets, each example is represented as a single-channel image with dimensions of $28 \times 28$ pixels, categorized into one of 10 classes. Additionally, the *Colored-MNIST(CMNIST)* [89] dataset, an extension of the original MNIST, introduces color into the digits of each example. Consequently, images in the Colored MNIST dataset are represented in three channels. *CIFAR* [92]: The *CIFAR-10* [92] dataset comprises 60,000 images, each with dimensions of $32 \times 32$ pixels and three color channels, distributed across 10 classes. This dataset includes 6,000 images per class and is partitioned into 50,000 training examples and 10,000 test examples. Similarly, the *CIFAR-100* [92] dataset shares the same image dimensions and structure as *CIFAR-10* but extends to 100 classes, with each class containing 600 images. Within each class, there are 500 training images and 100 test images. Moreover, *CIFAR-100* organizes its 100 classes into 20 superclasses, forming the *CIFAR-20 dataset* [92]. *CelebA* [84]: A face recognition dataset featuring 40 attributes such as gender and facial characteristics, comprising 162,770 training examples and 19,962 test examples. This study will focus on utilizing the *CelebA* [84] dataset primarily for gender classification tasks. ImageNet [93]: A large-scale image dataset which contains 1.2 million training samples across 1,000 categories.

*Adult Census Income (Adult)* [85] includes 48, 842 records with 14 attributes such as age, gender, education, marital status, etc. The classification task of this dataset is to predict if a person earns over $50K a year based on the census attributes. We then consider marital status as the sensitive feature that aim to unlearn in this study. *Diabetes* [86] includes 768 personal health records of females at least 21 years old with 8 attributes such as blood pressure, insulin level, age and etc. The classification task of this dataset is to predict if a person has diabetes. We then consider number of pregnancies as the sensitive feature that aim to unlearn in this study.

The IMDB movie reviews dataset [95] is widely used for binary sentiment analysis, where the task is to determine whether a review expresses a positive or negative sentiment. It comprises 50,000 movie reviews, each labeled as either positive or negative. In this study, we focus on unlearning the influence of specific sensitive features, particularly the names of celebrities. Each client's local dataset includes names of specific celebrities, which are treated as sensitive features for this analysis.

**Baselines** The baseline methods in this study:

*Baseline*: Original model before unlearning.

*Retrain*: In scenarios involving sensitive feature unlearning, the retrained model was simply trained using a dataset where Gaussian noise was applied to the unlearned feature region. This approach

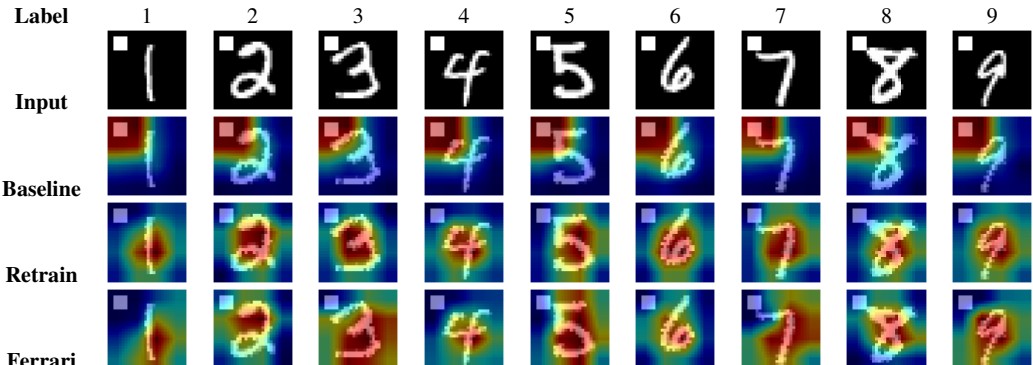

Figure 9: MNIST

may lead to performance deterioration, as discussed in Sec. 3.2. For backdoor feature unlearning scenarios, the retrained model was trained using the retain dataset $\mathcal{D}_r$, also referred to as the clean dataset. In biased feature unlearning scenarios, the retrained model was trained using a combination of 50% from each of the retain dataset $\mathcal{D}_r$ (bias dataset) and the unlearn client local dataset $\mathcal{D}_u$ (unbias dataset). This ensures fairness in the model's performance across both datasets.

*Fine-tune*: The baseline model is fine-tuned using the retained dataset $\mathcal{D}_r$ for 5 epochs.

*Class-Discriminative Pruning(FedCDP)* [65]: A FU framework that achieves class unlearning by utilizing Term Frequency-Inverse Document Frequency (TF-IDF) guided channel pruning, which selectively removes the most discriminative channels related to the target category and followed by fine-tuning without retraining from scratch.

*FedRecovery* [61]: A FU framework that achieves client unlearning by removing the influence of a client's data from the global model using a differentially private machine unlearning algorithm that leverages historical gradient submissions without the need for retraining.

## A.3   Attention Map

In this section, we provide additional results from attention map analysis based on GradCAM [96] for backdoor feature unlearning (refer to Sec. A.3.1) and biased feature unlearning (refer to Sec. A.3.2)

### A.3.1   Backdoor Feature Unlearning

Attention map analysis for backdoor samples across model iterations of baseline, retrain, and unlearn model using our proposed Ferrari method on MNIST (Fig. 9), FMNIST (Fig. 10), CIFAR-10 (Fig. 11), CIFAR-20 (Fig. 12) and CIFAR-100 (Fig. 13) datasets.

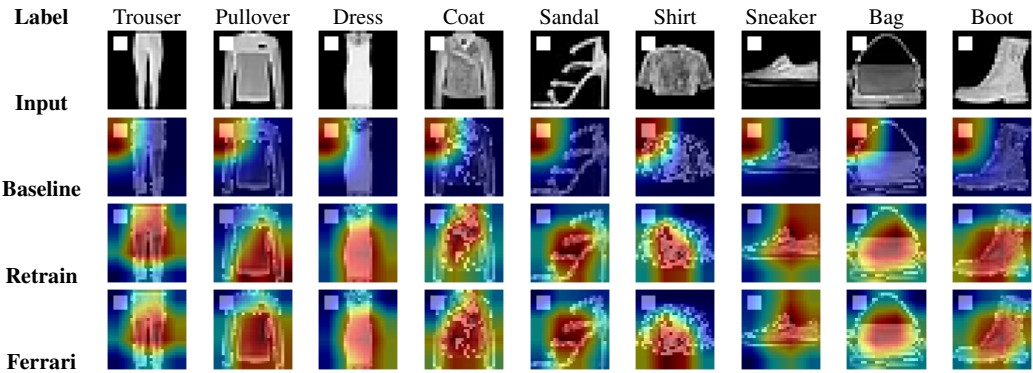

Figure 10: FMNIST

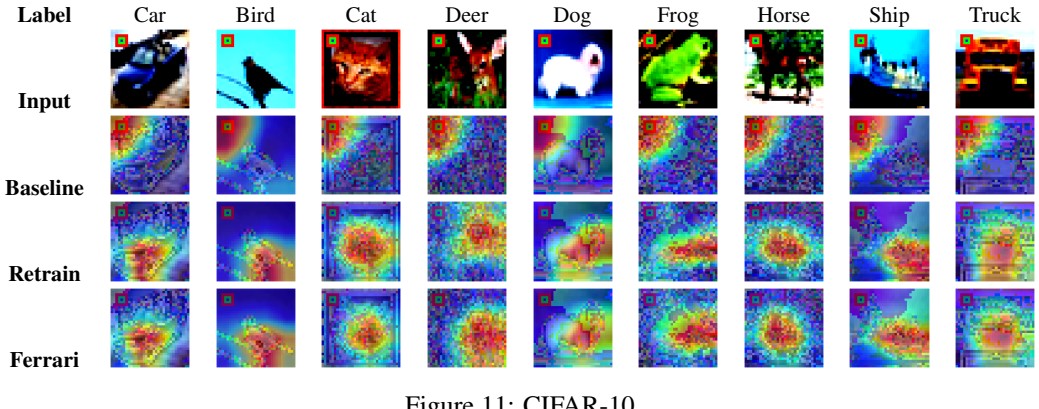

Figure 11: CIFAR-10

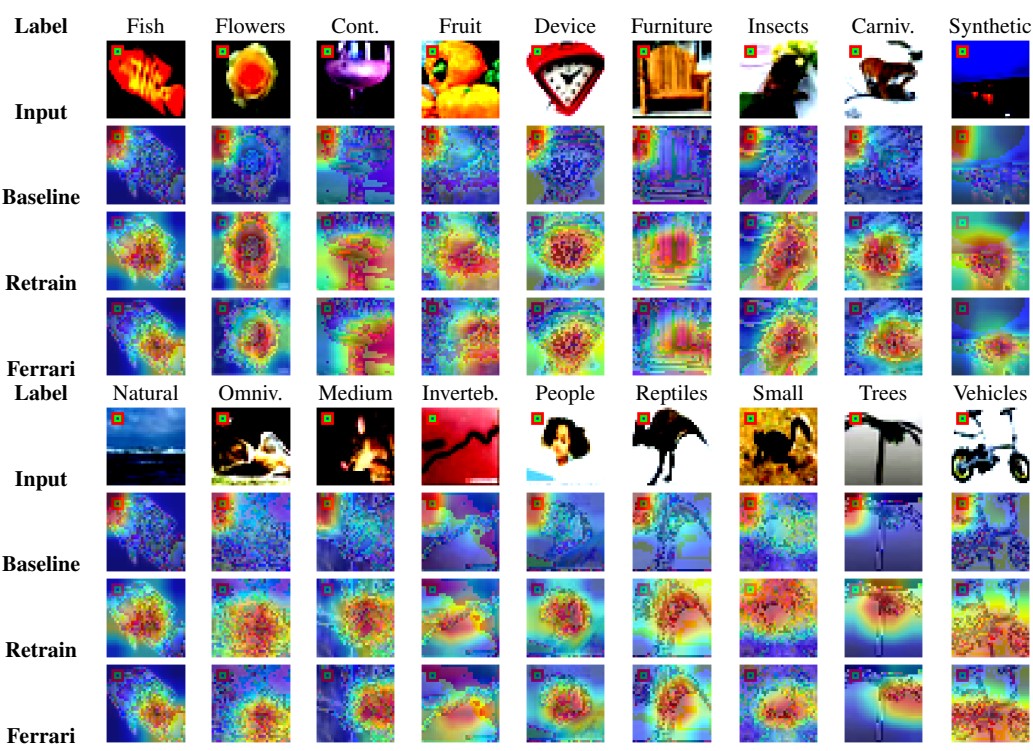

Figure 12: CIFAR-20

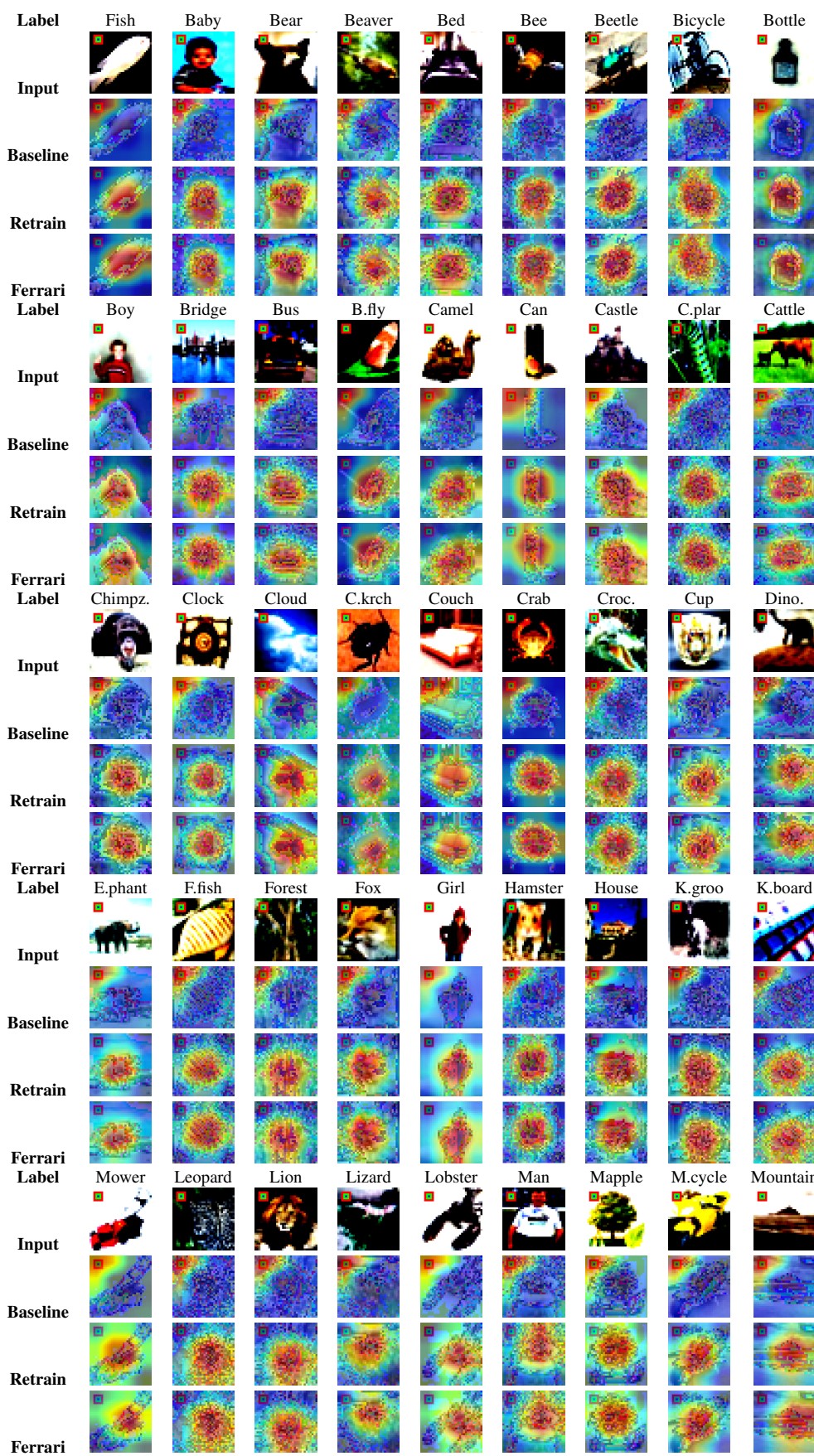

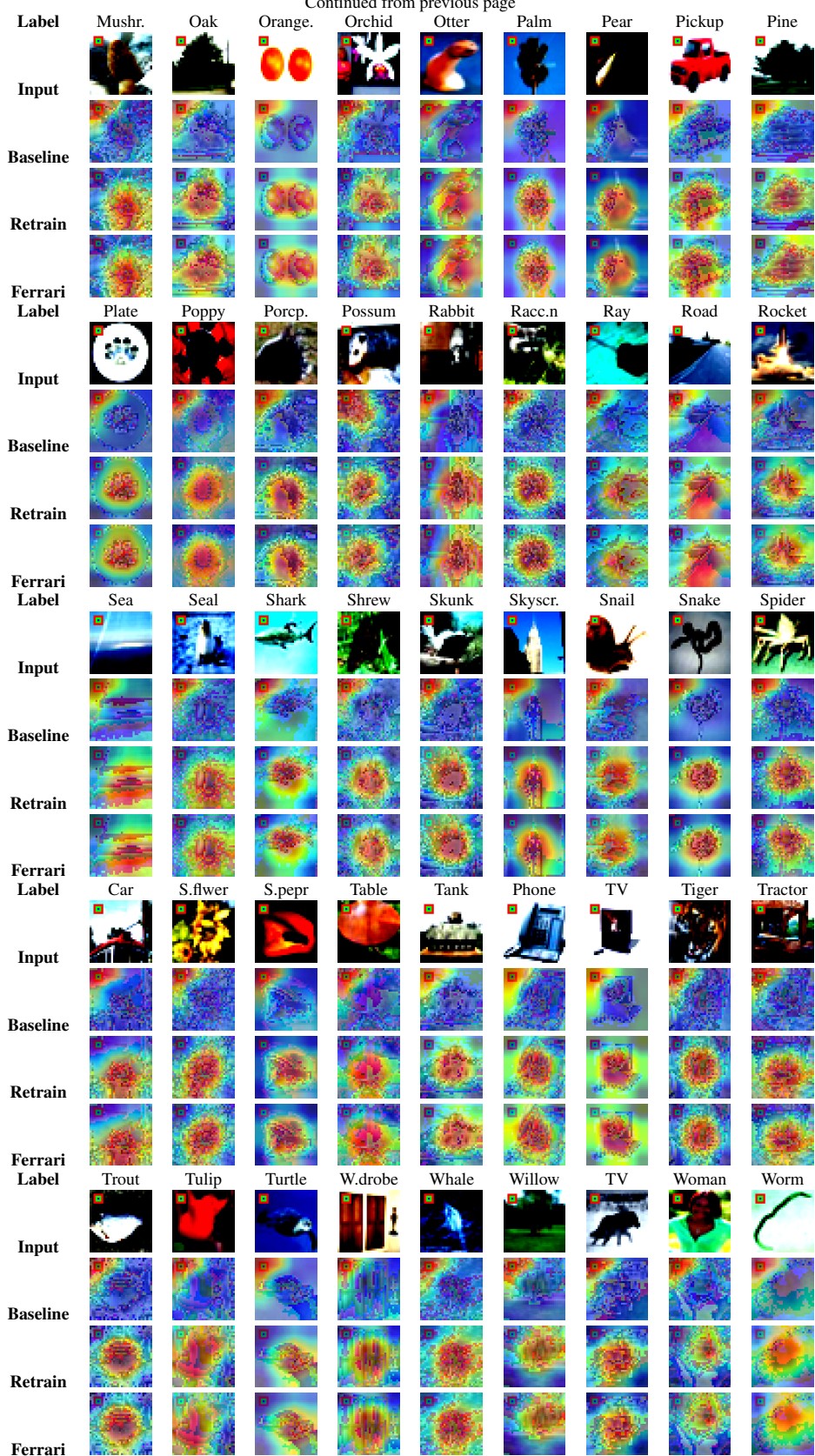

Figure 13: CIFAR-100

### A.3.2 Biased Feature Unlearning

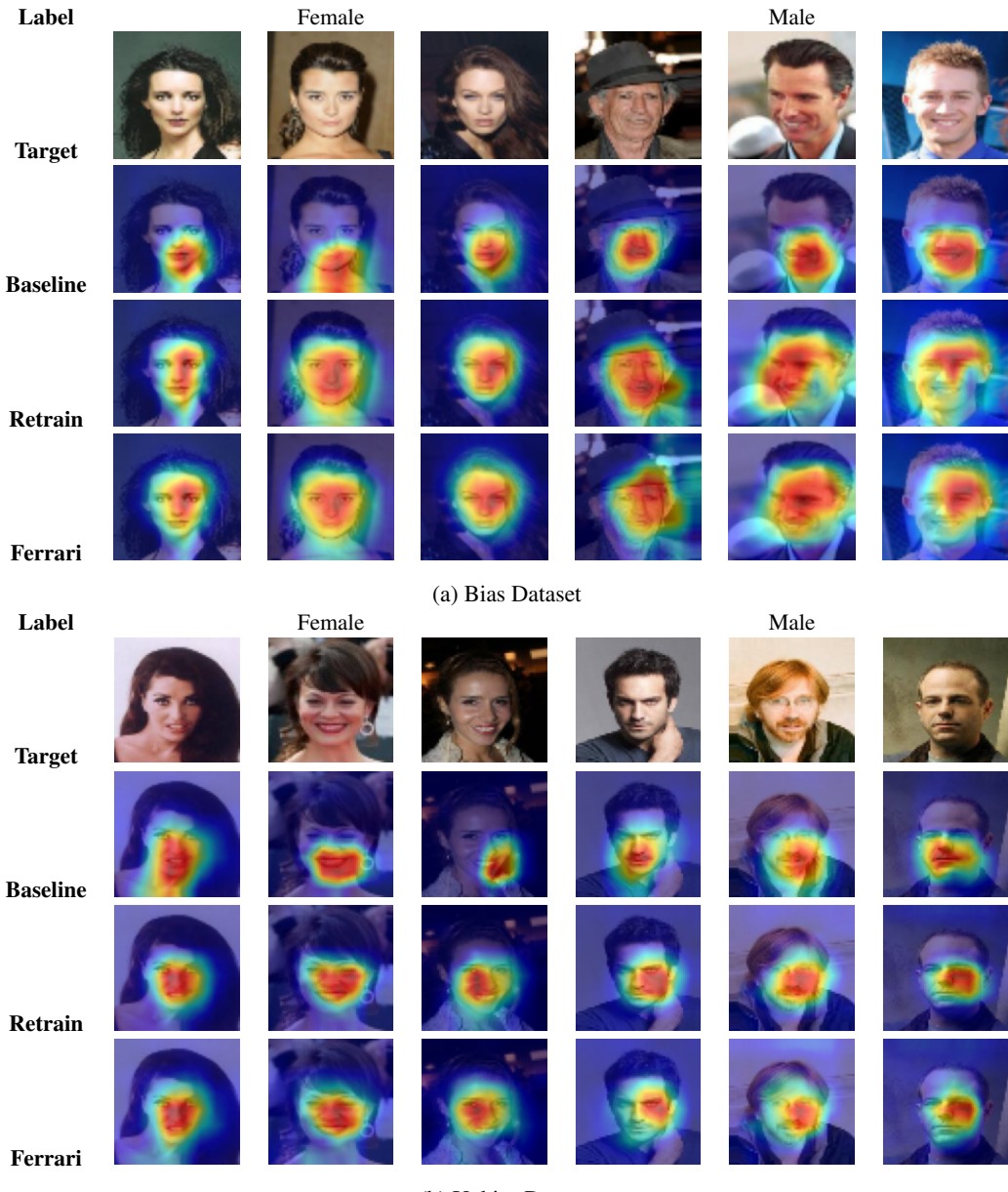

(a) Bias Dataset

(b) Unbias Dataset

Figure 14: Attention map analysis for bias and unbias samples across model iterations of baseline, retrain, and unlearn model using our proposed Ferrari to unlearn 'mouth' on CelebA dataset.

### A.4 Lipschitz and Non-Lipschitz Loss Analysis

In this section, we evaluate the Lipschitz loss function and its effectiveness in optimizing feature sensitivity, as described in Eq. 6. We also examine a variant without the denominator, termed the Non-Lipschitz loss, as illustrated in Fig. 15.

The results indicate that models optimized using the Non-Lipschitz loss exhibit fluctuations across batches. This is due to the unbounded nature of the optimization process, leading to useless models. Fig. 8a further illustrates this issue, showing instances of catastrophic forgetting.

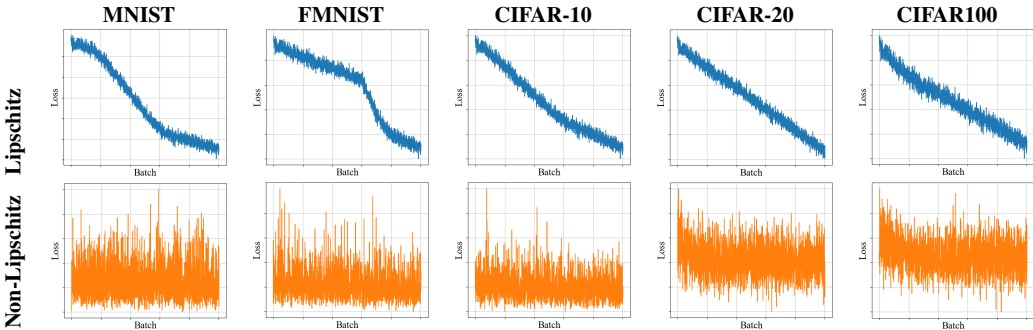

Figure 15: Lipschitz and Non-Lipschitz loss analysis on backdoor feature unlearning.

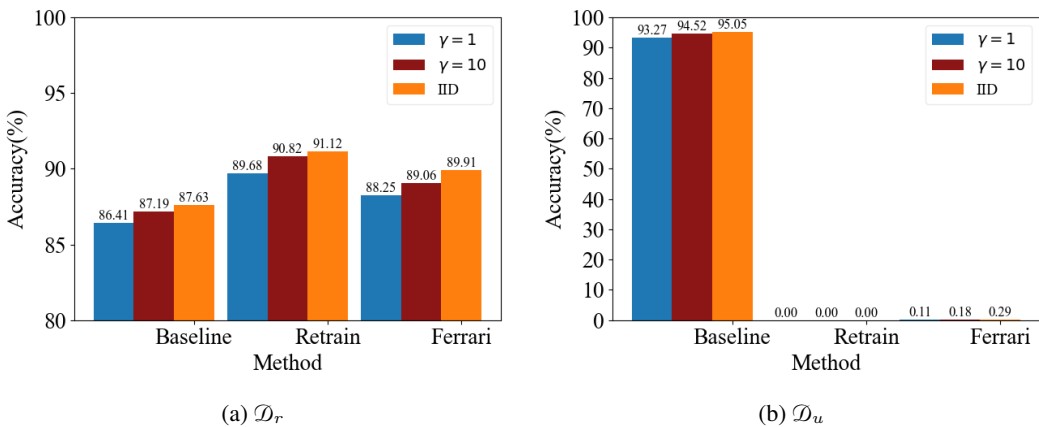

(a) $\mathcal{D}_r$                    (b) $\mathcal{D}_u$

Figure 16: Non-IID analysis on the CIFAR-10 dataset using our proposed Ferrari framework, compared to Baseline and Retrain methods, for retain client dataset $\mathcal{D}_r$ and unlearn client dataset $\mathcal{D}_u$ accuracy in backdoor feature unlearning.

Conversely, models optimized with the Lipschitz loss demonstrate a steady reduction in feature sensitivity over batches. This bounded optimization provided by Lipschitz bound helps in effectively unlearning target features while preserving model utility, as theoretically guaranteed (see Section Sec. 4.3).

### A.5   Non-IID Analysis

This section presents an analysis of the impact of Non-IID data on the performance of the proposed Ferrari framework compared to the Baseline and Retrain methods on the CIFAR-10 dataset. We focus on the accuracy of the retain client dataset ($\mathcal{D}_r$) and the unlearn client dataset ($\mathcal{D}_u$) in backdoor feature unlearning, as illustrated in Fig.16. To measure the extent of Non-IID, we used the $Dir(\gamma)$ distribution, where smaller values of $\gamma$ indicate more heterogeneous data.

The results show that the Ferrari framework significantly improves feature unlearning performance, with a drop of approximately 0.2% in $\mathcal{D}_u$ when $\gamma = 1$ compared to the IID scenario. Furthermore, the Ferrari framework maintains successfully the accuracy of $\mathcal{D}_r$ with only a slight decrease of about 2% compared to the Retrain method within the Non-IID scenario.

### A.6   Client Numbers Analysis

This section analyzes the impact of a large-scale FL environment, characterized by a large number of clients, on the performance of the proposed Ferrari framework compared to the Baseline and Retrain methods on the CIFAR-10 dataset. We focus on the accuracy of the retained client dataset ($\mathcal{D}_r$) and the unlearned client dataset ($\mathcal{D}_u$) in backdoor feature unlearning, as illustrated in Fig.17. The results indicate that the unlearning performance of our proposed Ferrari framework remains

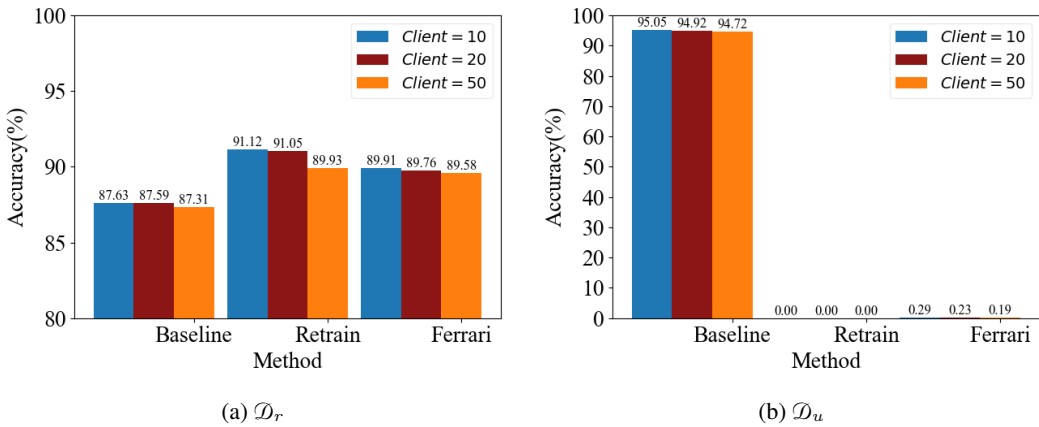

(a) $\mathscr{D}_r$            (b) $\mathscr{D}_u$

Figure 17: Scability analysis of client numbers on the CIFAR-10 dataset on the accuracy of retain client dataset $D_r$ and unlearn client dataset $D_u$

consistent, with no significant changes in the accuracy of both $\mathscr{D}_r$ and $\mathscr{D}_u$ as the number of clients increases. This finding further demonstrates the effectiveness of the Ferrari framework in large-scale FL environments.

