# OpenReview forum: "Ferrari: Federated Feature Unlearning via Optimizing Feature Sensitivity"
_NeurIPS.cc/2024/Conference — NeurIPS 2024 poster_

### Official Review · Reviewer_aV7Y · 2024-06-26

**Soundness:** 3
**Presentation:** 2
**Contribution:** 2
**Rating:** 6
**Confidence:** 3

**Summary:**

The paper introduces "Ferrari," a framework for federated feature unlearning that optimizes feature sensitivity to address the 'right to be forgotten' in machine learning. It allows clients to remove sensitive or biased features from a model without full retraining or other clients' data. Ferrari is efficient, practical, and theoretically proven to cause less utility loss than exact unlearning, with empirical results validating its effectiveness across various datasets.

**Strengths:**

1. Originality: Ferrari is a fresh take on federated learning, focusing on a new angle of data privacy.
2. Quality: The paper is well organized, with a thorough set of experiments.
3. Clarity: The presentation is clear. The authors have successfully demarcated the problem, the proposed solution, and the results in a manner that is easy to follow. The use of figures and tables enhances the readability and helps in quickly grasping complex concepts.
4. Significance: This work is important for machine learning, especially for keeping data private. It helps with rules and doing AI the right way

**Weaknesses:**

The paper demonstrates a commendable effort in advancing federated learning with a focus on feature unlearning. However, there are a few areas where the work could be further strengthened:
1. The format of the article needs to be re-checked, for example, there are some formulas after the punctuation is set incorrectly
2. Since feature unlearning is important, it would be helpful to see how well Ferrari stands up to attacks that try to bring back forgotten features.
3. More details on using Ferrari in the real world would be useful, considering things like network delays, client availability, and the extra work on client devices.

**Questions:**

1. How does Ferrari defend against potential adversarial attacks aimed at reintroducing unlearned features? Is there a plan to evaluate and enhance its robustness in this area?
2. What is the computational complexity of the Ferrari framework, and how does it compare to existing federated learning methods in terms of efficiency and resource usage?
3. The paper relies on certain theoretical assumptions for its analysis. How do these assumptions hold in real-world scenarios, and are there plans to validate them empirically?
4. What open questions or areas for future work do the authors identify based on their research? How do they envision Ferrari evolving to address these challenges?

**Limitations:**

The authors have commendably addressed the limitations of their work with a clear and honest discussion, enhancing the research's integrity and transparency. To improve further, they should explore how their assumptions might affect real-world applications and suggest ways to handle any negative effects. Also, a deeper look into societal impacts, including unintended consequences, would be beneficial. Addressing these will help provide a fuller understanding of the broader impact of their work.

---

> ### Author Rebuttal · Authors · 2024-08-07
>
> `` 1. Defend Adversarial Attack ``
>
> Our framework effectively defends against Model Inversion Attacks (MIA) [1] by preventing the reconstruction of the target unlearn features. As shown in Tab 3 in main paper, the attack success rate with our framework is similar to that of the naive retrain method, indicating that unlearned features are protected. In contrast, the baseline model before unlearning has a 90\% attack success rate, highlighting severe data privacy risks. For other baselines like FedCDP and FedRecovery, the success rate is about 80\%. Our Ferrari framework reduces this rate to around 50\%, significantly enhancing privacy. Tab 4 shows that our framework prevents reconstruction of the target unlearned feature (e.g., the mouth), whereas the baseline model does not.
>
> Ferrari effectively defends against MIA [1, 2] by preventing the reconstruction of the target unlearn features. As shown in Tab 3 (main paper), the attack success rate with our framework is similar to that of the naive retrain method, indicating that unlearned features are protected. In contrast, the baseline model before unlearning has a 90\% attack success rate, highlighting severe data privacy risks. For other baselines like FedCDP and FedRecovery, the success rate is about 80\%. Ferrari reduces this rate to around 50\%, significantly enhancing privacy. Tab 4 shows that our framework prevents reconstruction of the target unlearned feature (e.g., the mouth), whereas the baseline model does not.
>
> **Ref:**
>
> #### _[1] Fredrikson et al, “Model inversion attacks that exploit confidence information and basic countermeasures,” CCS ’15, (New York, NY, USA), p. 1322–1333, Association for Computing Machinery, 2015_
>
> #### _[2] Y. Zhang et al, “The secret revealer: Generative model inversion attacks against deep neural networks,” in 2020 IEEE/CVF Conference on Computer Vision and Pattern Recognition (CVPR), pp. 250–258, 2020_
>
> `` 2. Computational Complexity ``
>
> We discussed time efficiency in Section 5.4 (main paper). Additionally, we present the computational complexity of our proposed Ferrari framework using FLOPs metrics, as shown in Fig 5 of the rebuttal PDF. Our Ferrari framework exhibits the lowest FLOPs, achieved by accessing only the local dataset of the unlearned client and minimizing feature sensitivity within a single epoch.
>
> `` 3. Theoretical Assumption ``
>
> We explain more on Assumption 1-2.
>
> - First, Assumption 1 elucidates that the utility loss associated with a perturbation norm less than $C $ is smaller than the utility loss when the perturbation norm is greater than $C$. This assumption is logical, as larger perturbations would naturally lead to greater utility loss.
>
> - Second, we can also extend the Theorem 1 to the case of the non-zero training loss. This is because
>
> \begin{equation}
>    \mathbb{E}\_{(x,y)\in \mathcal{D}} \left( \|\| f\_{\theta^*}(x) - y \|\| + \lambda \mathbb{E}\_{\|\| \delta_F \|\| \geq \lambda} \frac{\|\|f\_\theta(x) - f\_{\theta^*} (x + \delta\_{F})\|\|_2}{\|\|\delta\_{F}\|\|_2} \right) \\
> \end{equation}
>
> \begin{equation}
>   = \mathbb{E}_{(x,y)\in \mathcal{D}} \mathcal{l}\|\| f\_{\theta^*}(x) - y \|\| + \lambda \mathbb{E}\_{(x,y)\in \mathcal{D}} \mathbb{E}\_{\|\| \delta_F \|\| \geq \lambda} \frac{\|\|f\_\theta(x) - f\_{\theta^*} (x + \delta\_{F})\|\|_2}{\|\|\delta\_{F}\|\|_2}
> \end{equation}
>
> \begin{equation}
>   \leq \mathbb{E}_{(x,y)\in \mathcal{D}} \mathcal{l} \|\| f\_{\theta^*}(x) - y \|\| + \mathbb{E}\_{(x,y)\in \mathcal{D}} \mathbb{E}\_{\|\| \delta_F \|\| \geq \lambda} \|\|f\_\theta(x) - f\_{\theta^*} (x + \delta\_{F})\|\|_2
> \end{equation}
>
> \begin{equation}
>   \leq  \mathbb{E}\_{(x,y)\in \mathcal{D}} \mathbb{E}\_{\|\| \delta_F \|\| \geq \lambda} \|\|y - f\_{\theta^*} (x + \delta\_{F})\|\|_2
> \end{equation}
>
> which obtain Eq. (13) in appendix (noted that $\|\delta_\mathcal{F}\| \geq \frac{1}{\lambda}$ in original appendix should be $\|\delta_\mathcal{F}\| \geq {\lambda}$).
>
> `` 4. Future Work ``
>
> We identify key areas for future research on federated feature unlearning with Ferrari:
>
> * **Dataset Access:** Explore methods that require less than 70\% of the unlearning client's dataset for effectiveness.
>
> * **Model Types:** Investigate the approach's applicability to non-classification models, such as generative models.
>
> * **Advanced Perturbation:** Develop advanced perturbation techniques to improve unlearning effectiveness.
>
> * **Data and Model Diversity:** Support a wider range of data types and models.
>
> * **Privacy Integration:** Integrate federated feature unlearning with other privacy-preserving methods for enhanced data protection.
>
> Future work should focus on minimizing dataset requirements, broadening model applicability, and improving robustness and privacy guarantees.

---

### Official Review · Reviewer_mo2d · 2024-07-10

**Soundness:** 2
**Presentation:** 3
**Contribution:** 2
**Rating:** 5
**Confidence:** 4

**Summary:**

This paper studies the feature unlearning problem in the context of federated learning. The key idea is to interpret feature unlearning into model sensitivity at specific features. The proposed method which is called Ferrari is evaluated on different tasks which involves sensitive, backdoor, and biased feature unlearning.

**Strengths:**

Federated unlearning is not something new in research community but focusing on feature unlearning is interesting. I agree with the authors that evaluating exact re-training for feature unlearning is a challenge to this research topic. The key idea of this paper is also straightforward and well-presented.

**Weaknesses:**

1. The challenges of this paper are not well presented.
2. The sensitivity reduction is not a proper solution of federated feature unlearning.
3. Important experiments are missing.

**Questions:**

Major concerns:
1. Suppose client i and client j have similar data point x. If client i claims to forget some "key features" while client j does not, the global model still has a chance to learn the "key features" without breaking the unlearning principle. Thus, using sensitivity reduction on the global model does not make sense. Similar to sample-wise unlearning in classification, you cannot simply interpret this form of unlearning as misclassifying this specific sample. I want to highlight this concern as I understand you are not the only one thinking in this way.
2. Now that you provide examples of unlearning users' sensitive information like name and address, I would expect you to have experiments on a text dataset as strong evidence, which is so far lacking in the paper.

Other minor questions:
1. Please check the reference, [10] and [11] are repeated.
2. After checking the paper [16], where they consider the malicious clients, I begin to understand “datasets contain backdoor triggers” you used. To be clearer, you’d better refer to local clients’ data.
3. Regarding the third scenario, imbalance is only a possible source of bias in terms of some specific fairness metrics. Please be careful when using these concepts.
4. The presented first challenge (line 32 on page 1) is interesting, while the second one should exist in client, class, or sample level. If so, you should put some efforts here to explain why this challenge remains by adapting existing strategies.
5. Following 4, you claimed a different challenge (challenge 2) in the first contribution.
6. Your method Ferrari is short for Federated Feature Unlearning, which is also the name of the learning problem. You’d better use different names.
7. Repeated phrases in line 105 on page 3.
8. Below Eq. 2, the unlearned feature set f is not indexed by sample i but this is true in the algorithm.

**Limitations:**

As I pointed out in Weaknesses & Questions, the technique may have some flaws in addressing the Federated Feature Unlearning problem. The experiments are primarily conducted on image samples, which cannot confidently prove the idea.

---

> ### Author Rebuttal · Authors · 2024-08-07
>
> `` 1. Challenge of paper ``
>
> Thank you for your suggestions. We clarify the second challenge from the original manuscript as follows:
>
> Firstly, previous feature unlearning methods [1, 2] in centralized settings are impractical for Federated Learning (FL), as they require access to all datasets and client participation. In FL, retained clients may refuse to participate due to privacy or computational concerns. Therefore, the most practical approach requires only the participation of the unlearn client.
>
> Secondly, existing approaches to Federated Unlearning primarily focus on unlearning at the client, class, or sample level. However, these methods do not address the challenge of feature unlearning in federated settings. They are limited to isolated data points and cannot manage the removal of specific features across multiple data points. Our results highlight that benchmark Federated Unlearning frameworks like FedCDP [3] and FedRecovery [4] struggle with effective feature unlearning in FL environments.
>
> **Ref:**
>
> #### _[1] Warnecke et. al, “Machine unlearning of features and labels,” in Proc. of the 30th Network and Distributed System Security (NDSS), 2023_
>
> #### [2] _Guo et. al, "Efficient attribute unlearning: Towards selective removal of input attributes from feature representations." arXiv preprint arXiv:2202.13295 (2022)._
>
> #### [3] _Wang et. al, “Federated unlearning via class-discriminative pruning,” in Proceedings of the ACM Web Conference 2022, WWW ’22, (New York, NY, USA)_
>
> #### [4] _Zhang et. al, “Fedrecovery: Differentially private machine unlearning for federated learning frameworks,” IEEE Transactions on Information Forensics and Security, vol. 18, pp. 4732–4746, 2023_
>
> `` 2. Unlearning Concern ``
>
> Thanks for you comments. First, **federated feature unlearning happens at the end of the federated training**. When the federated training ends, unlearned client implements the unlearning operation to obtain the unlearned model $\theta_u$ finally the server replace the original model with $\theta_u$.
>
> Second, in the case of IID data, **it is valuable to unlearn features**. For instance, a client (Alice) may want to remove the "marital-status" feature from adult data. Even if other clients have the same feature, Alice still needs to remove this information (marriage status).
>
> Third, the experimental results in Tabs 2 and 3 (main paper) demonstrate that **the proposed Ferrari can successfully remove the feature of one client, even in the IID setting**.
>
> Finally, beyond the IID scenario, our framework remains effective for feature unlearning in Non-IID scenarios, as illustrated in Fig 3 of the rebuttal PDF. The results show that the Ferrari significantly improves feature unlearning performance, with only a 0.2\% drop in $D_u$ when $\gamma = 1$ compared to the IID scenario. Furthermore, considering the backdoor unlearning scenario, where backdoor data from one client has a different distribution compared to the normal data.
>
> `` 3. Text Dataset ``
>
> Thank you for your suggestion. As suggested, we conducted text classification using the IMDB movie reviews dataset [1] with a BERT model to analyze feature unlearning by removing the influence of sensitive features, specifically celebrity names. Each client’s local dataset includes only a specific name of a celebrity. As to Table below, Ferrari framework proves highly effective, maintaining high model accuracy while achieving the lowest feature sensitivity and MIA attack success rate, thus preventing the reconstruction of sensitive features and enhancing data privacy.
>
> > - Model Utility Analysis:
>
> | **Scenarios**  | **Datasets** | **Unlearn Feature** | **Baseline**       | **Retrain**        | **Fine-tune**      | **FedCDP**         | **FedRecovery**    | **Ours** |
> |----------------|--------------|---------------------|--------------------|--------------------|--------------------|--------------------|--------------------|-----------------------|
> | Sensitive | IMDB     | Names               | 91.39 ± 1.57       | 83.27 ± 2.05       | 72.15 ± 1.92       | 48.36 ± 2.79       | 37.93 ± 2.84       | **89.15 ± 1.32**      |
>
>
> > - Feature Sensitivity Analysis:
>
> | **Scenario**  | **Datasets** | **Unlearn Feature** | **Baseline**          | **Retrain**           | **Fine-tune**         | **FedCDP**            | **FedRecovery**       | **Ours** |
> |---------------|--------------|---------------------|------------------------|------------------------|------------------------|------------------------|------------------------|------------------------|
> | Sensitive | IMDB     | Names               | 0.85 ± 1.07×10⁻²       | 0.07 ± 5.38×10⁻⁴       | 0.74 ± 3.81×10⁻²       | 0.81 ± 3.27×10⁻²       | 0.78 ± 2.41×10⁻²       | **0.08 ± 1.32×10⁻⁴**   |
>
> > - Model Inversion Attack Analysis:
>
> | **Scenario**   | **Datasets** | **Unlearn Feature** | **Baseline**       | **Retrain**        | **Fine-tune**      | **FedCDP**         | **FedRecovery**    | **Ours** |
> |----------------|--------------|---------------------|--------------------|--------------------|--------------------|--------------------|--------------------|-----------------------|
> | Sensitive  | IMDB     | Names               | 90.28 ± 2.49       | 40.29 ± 1.59       | 86.74 ± 3.81       | 83.67 ± 4.59       | 80.95 ± 3.51       | **43.75 ± 1.86**      |
>
> **Ref:**
>
> #### _[1] https://www.kaggle.com/datasets/lakshmi25npathi/imdb-dataset-of-50k-movie-reviews_
>
> `` 5. Minor Issues ``
>
> - The repetitive references have been removed in our manuscript, preventing confusion for the reader.
>
> - Thank you for your suggestion. We will make the necessary changes to enhance the clarity of our work.
>
> - Thank you for your suggestion. We will request permission from TPC to update the short title of this paper. If approve, we will update the title accordingly.
>
> - Repeated phrases in line 105 has been removed to improve the readability.
>
> - We corrected the unlearned feature set $F$ to $F_i$ in Equation 2 (main paper)

---

> > ### Comment · Reviewer_mo2d · 2024-08-13
> > **Thanks for your response**
> >
> > Regarding my first concern, I was attempting to question a scenario when two clients (i and j) happen to have a pair of similar data point, e.g, x1 from client i and x2 from j which are very similar on features. In this sense, they may be equally important for model training. But if client i requires to unlearn x1 while client j does not requires to unlearn x2, then how will global model response?
> >
> > As what I have commented before, using sensitivity reduction on the global model would simply encourage sensitivity reduction on x1, just similar to sample-wise unlearning in classification where you cannot simply interpret this form of unlearning as misclassifying this specific sample. For example, if x1 and x2 are supported vectors of the classifier, unlearning x1 will not necessarily need to change the decision boundary because x2 is still there.
> >
> > From my understanding, unlearning a sample equals to unseeing it during training instead of unfitting it. I would like to highlight this case as some machine unlearning players may ignore it.

---

> ### Author Response · Authors · 2024-08-13
> **Thank you for your thoughtful feedback**
>
> ``1.  Global Model Response``
>
> We sincerely apologize for any confusion in our previous communication.
>
> In scenarios with IID, unlearning a feature from a target client will inadvertently lead to unlearning similar features in other clients within the global model. For example, we use a face classification task to assess if unlearning _'mouth'_ feature for one client impacts the model's sensitivity to the _'mouth'_ feature in the other nine clients.
>
> - (i) In Table 1, the feature sensitivity for unlearned client ($D_u$) and retained client ($D_r$) is significantly reduced (e.g., dropping below 0.11), compared to values > 0.95 before unlearning.
>
> - (ii) In Table 2, the successful Membership Inference Attack (MIA) rate for features from unlearned client ($D_u$) and retained client ($D_r$) decreases to approx. 50%, whereas it was > 80% before unlearning.
>
> >Table 1: Feature sensitivity analysis in FL including 10 clients under IID, a *small* feature sensitivity indicates good unlearning effect
> | **Dataset** | **Unlearn Feature** | **$\mathcal{D}_u$ (Before Unlearning)** | **$\mathcal{D}_r$ (Before Unlearning)** | **$\mathcal{D}_u$ (Retrain)** | **$\mathcal{D}_r$ (Retrain)** | **$\mathcal{D}_u$ (Ours)** | **$\mathcal{D}_r$ (Ours)** |
> |-------------|---------------------|-----------------------------------------|-----------------------------------------|-------------------------------|-------------------------------|----------------------------|----------------------------|
> | CelebA      | Mouth               | 0.96 ± 1.41×10⁻²                        | 0.95 ± 1.63×10⁻²                        | 0.07 ± 8.06×10⁻⁴               | 0.08 ± 7.51×10⁻⁴               | 0.09 ± 3.04×10⁻⁴            | 0.11 ± 3.62×10⁻⁴            |
>
> >Table 2: Model inversion attack in FL including 10 clients under IID, a *small* MIA indicates good unlearning effect
> | **Dataset** | **Unlearn Feature** | **$\mathcal{D}_u$ (Before Unlearning)** | **$\mathcal{D}_r$ (Before Unlearning)** | **$\mathcal{D}_u$ (Retrain)** | **$\mathcal{D}_r$ (Retrain)** | **$\mathcal{D}_u$ (Ours)** | **$\mathcal{D}_r$ (Ours)** |
> |-------------|---------------------|-----------------------------------------|-----------------------------------------|-------------------------------|-------------------------------|----------------------------|----------------------------|
> | CelebA      | Mouth               | 84.36 ± 3.22                            | 84.17 ± 2.93                            | 47.52 ± 1.04                   | 48.36 ± 1.39                   | 51.28 ± 2.41                | 52.46 ± 2.76                |
>
> ``2. Global Sensitivity Reduction ``
>
> Thank you, we appreciate the opportunity to clarify further.
>
> Firstly, under **IID setting** in federated learning (FL), we recognize the concern regarding the removal of sensitive features from the global model, especially when two clients have similar or identical features and only one requests unlearning. However, it is important to note that retaining such a sensitive feature in the global model may lead to potential violations of regulations such as GDPR. Thus, we argue that removing the feature from the global model is a reasonable and necessary step to ensure compliance. Also, as demonstrated in Table 1 (main paper), the removal of sensitive features for unlearned clients **does not significantly diminish the model's utility for the remaining clients**. Thus, under **IID setting**, our unlearning strategy offers a practical and acceptable solution that balances compliance with maintaining the effectiveness of the global model.
>
> Secondly, unlearning in other scenarios within FL, such as **Non-IID**, **backdoor unlearning**, and **bias unlearning**, our proposal holds particular value. In Non-IID and backdoor, individual clients often possess personalized features that are unique to them, unlike the shared features in the IID setting. For bias unlearning, the shared objective across all clients is to remove bias from the global model. As shown in Sections 5.3.2 - 5.3.3 (main paper), *Ferrari* **performs effectively** in unlearning bias and backdoor features; as well as in Non-IID setting (please refer Fig. 3 in the rebuttal PDF).
>
> ``3. Unseeing as unlearning``
>
> Thank you. **The feature unlearning herein is equal to "unseeing" it during training instead of unfitting it**. According to [1], exact feature unlearning can be similarly defined as: $\theta_u \sim \theta_r$, where $\theta_u$ is the unlearned model and $\theta_r$ is the retrained model using data $x+\delta_{\mathcal{F}}$. Our proposed aims to optimize:
>
> \begin{equation}
> \min_\theta \mathbb{E}_{(x,y) \in \mathcal{D}_u} \frac{1}{N} \sum\_{i=1}^N \frac{\|\|f\_\theta(x) - f\_\theta (x + \delta\_{F,i})\|\|_2}{\|\|\delta\_{F,i}\|\|_2}
> \end{equation}
>
> This objective drives the unlearned model to converge toward the retrained model, effectively making the unlearned model **unsee** the feature.
>
> [1] Bourtoule et al., “Machine unlearning,” in 2021 IEEE SP.

---

### Official Review · Reviewer_49hC · 2024-07-11

**Soundness:** 4
**Presentation:** 4
**Contribution:** 4
**Rating:** 8
**Confidence:** 5

**Summary:**

This paper introduces a novel framework named Ferrari, addressing two challenges for federated feature unlearning. Ferrari leverages Lipschitz continuity to minimize the feature sensitivity directly during the local unlearning process, demonstrating several key advantages, such as eliminating the requirement of other clients' participation, supporting various unlearning scenarios, and achieving less model utility loss. Both the theoretical proof and empirical results demonstrate the effectiveness of Ferrari in a variety of unlearning settings compared with other baselines.

**Strengths:**

1. The authors propose a novel method for federated feature unlearning, demonstrating several key advantages, such as eliminating the requirement of other clients' participation, supporting various unlearning scenarios, and achieving less model utility loss. This approach somehow addresses two challenges during FL unlearning and makes some contributions to this particular field of research.
2. The Ferrari method is applicable to a wide range of unlearning tasks and shows potential for integration with other unlearning techniques in the future.
3. The authors have provided both theoretical proofs and empirical results to demonstrate the soundness and effectiveness of the proposed approach.
4. The paper is clearly written and well-organized. It is easy to follow the authors' ideas and understand their ideas, including the problem formulation, challenges, and the proposed methods. The authors use clear figure and algorithm, i.e., Figure 2 and Algorithm 1, to show the procedure of their method. The notations and experimental results are clear and easy to read.
5. The authors have conducted extensive experiments to demonstrate the effectiveness of Ferrari in a variety of unlearning settings and datasets compared with several baselines.

**Weaknesses:**

1. The authors should have provided more analysis on Non-Lipschitz besides the proofs, especially in the Ablation Studies section.
2. Ferrari's performance heavily relies on the number of unlearned datasets, as shown in Figure 7 (b). In a real-world scenario, the unlearning client may ask to unlearn all the datasets, not just a small portion.
3.  Minor issues: Gaussian Noise and the Number of Unlearned Datasets should be hyper-parameter tuning, not ablation studies. All the
single quotes and double quotes in the paper are not in the correct formats.

**Questions:**

1. What does the dotted and solid line in Figure 7 mean?
2.  How do you get the results of Figure 6? For me, if only retrain or fine-tune one specific client, the runtimes should be similar as Ferrari.

**Limitations:**

See the second point in weaknesses.

---

> ### Author Rebuttal · Authors · 2024-08-07
>
> `` 1. Non-Lipschitz Analysis ``
>
> We evaluate the Lipschitz loss function for optimizing feature sensitivity (Equation 6) and compare it with a variant lacking the denominator, termed the Non-Lipschitz loss (see Fig 1 of the rebuttal PDF). Results show that models using the Non-Lipschitz loss exhibit batch fluctuations and ineffective performance due to unbounded optimization, leading to catastrophic forgetting (Fig 7(c) in main paper). In contrast, models optimized with the Lipschitz loss achieve a steady reduction in feature sensitivity, effectively unlearning target features while maintaining model utility, as guaranteed theoretically (Section 4.3 of main paper)
>
> `` 2. Figure 7 Clarification ``
>
> In Fig 7, the **solid line** represents the retain client dataset $D_r$ while the **dotted line** represents the unlearn client local dataset $D_u$ in Federated Learning scenario.
>
> `` 3. Figure 6 Clarification ``
>
> The retrain model's runtime was measured by training the global model from scratch for 20-25 epochs until convergence, while the fine-tune model involved 5 epochs of fine-tuning the baseline model. In comparison, the Ferrari framework achieved local feature unlearning in just one epoch, resulting in significantly faster runtime than both the retrain and fine-tune methods.
>
> `` 4. Minor Issue ``
>
> To address space constraints, we combined the ablation study and hyper-parameter analysis into Section 5.5, renamed "Ablation and Hyper-parameter Analysis." This section separately discusses the Non-Lipschitz component and the Gaussian noise level and number of unlearned datasets. Additionally, we will correct all single and double quotes to the proper format.
>
> `` 5. Figure 7b Clarification ``
>
> Sorry for the confusion. Actually, Fig 7(b) intention is to demonstrate that our framework requires only at least 70\% of the unlearned client's local dataset to achieve optimal performance comparable to using the entire dataset. This highlights the framework's robustness and practicality, as it remains nearly optimal even when only 70\% of the dataset is available due to data loss.

---

> > ### Comment · Reviewer_49hC · 2024-08-12
> > **Thank you for your responses.**
> >
> > I have read the authors' responses. Most of my concerns have been addressed. I will keep my score as 8 Strong Accept.

---

> ### Author Response · Authors · 2024-08-12
> **Appreciation for upholding Strong Accept and supporting our work**
>
> We sincerely appreciate your thoughtful feedback and for upholding the **8 = Strong Accept** score. Your insightful suggestions have greatly improved the clarity and overall quality of our work. Thank you for your valuable support.

---

### Official Review · Reviewer_vD3D · 2024-07-11

**Soundness:** 2
**Presentation:** 3
**Contribution:** 3
**Rating:** 5
**Confidence:** 5

**Summary:**

The rise of Federated Learning (FL) has led to the importance of the 'right to be forgotten', inspiring a need for Federated Unlearning (FU). Feature unlearning, particularly for removing sensitive, backdoor, and biased features, has attracted significant interest. However, current methods relying on the influence function are not suitable for FL since they require other clients' involvement, and there is a lack of assessment regarding the feature unlearning's effectiveness. To tackle these problems, the paper introduces a novel concept, feature sensitivity, derived from Lipschitz continuity, as a metric to evaluate feature unlearning efficiency, quantifying the impact of changes in input features on the model's output. A new federated feature unlearning framework named Ferrari is then proposed, which aims to minimize feature sensitivity. Thorough experimental evidence and theoretical analysis validate Ferrari's effectiveness in various feature unlearning cases, including those involving sensitive, backdoor, and biased features.

**Strengths:**

+ This work offers a new FU framework that aims to minimize feature sensitivity, thereby achieving efficient unlearning from the perspective of features.
+ This paper is the first to study challenging feature unlearning in the context of FL.
+ Comprehensive and thorough evaluation.

**Weaknesses:**

- Further clarification is needed on the motivation for feature unlearning in FL.
- The rationality of the designed feature unlearning evaluation metric remains to be discussed.
- Further evaluation is needed in non-IID data scenarios.
- The utility loss loss seems inconsistent with the numerical results.
- The chosen baseline does not seem to be tailor-made for the evaluation task, which may lead to unfair comparisons.

**Questions:**

- Further clarification is needed on the motivation for feature unlearning in FL. In the case of IID data, where clients share similar features, a single user's request for feature unlearning might face challenges in fully eliminating targeted features from the global model, given that the model has already learned similar patterns from other clients. While data erasure seems to provide a higher level of user safety, it is crucial to understand why users might opt for feature erasure instead.

- In the context of Federated Unlearning (FU), privacy assurance takes precedence over utility loss analysis. Can the authors present a privacy analysis or provide relevant proof demonstrating the privacy guarantees of feature unlearning in the FU setting?

- The paper discusses the utility loss of Ferrari; however, paradoxically, Table 1 reveals that Ferrari's utility is superior to retraining in some instances and even surpasses the baseline in specific cases. It would be beneficial if the authors could offer more insight to further explain this unexpected outcome.

- Given the potential challenges posed by the non-IID data setting, it is essential to investigate Ferrari's performance under such conditions. Can the authors provide an exploration of Ferrari's effectiveness in a non-IID data scenario?

- Concerns exist about Ferrari's privacy performance, as although it outperforms other baselines in defending against MIA, it still presents a relatively high probability of successful MIA attacks. This raises questions about users' willingness to use Ferrari for feature unlearning. Would the authors consider integrating additional techniques to enhance its privacy performance further?

- Based on the observed outcomes from the MIA, doubts arise about the adequacy of the proposed validation metric for evaluating feature unlearning performance. It appears that the metric may not accurately reflect the effectiveness of the feature unlearning process. Therefore, it would be valuable if the authors could delve deeper into analyzing the reasoning behind and the correctness of the designed evaluation metric, to either validate its reliability or suggest improvements for a more accurate assessment of feature unlearning.

- Considering that the selected baselines such as FedCDP and FedRecovery are not tailor-made for feature unlearning, whether it is fair to compare directly with Ferrari needs further clarification.

**Limitations:**

Please kindly refer to the above comments in the Questions part.

---

> ### Author Rebuttal · Authors · 2024-08-07
>
> `` 1. Motivation ``
>
> Thank you for raising this important question about the motivation for feature unlearning in Federated Learning (FL). In the case of IID data, where clients may share similar features, it is indeed valuable to consider the need for unlearning specific features. For example, a client may request the removal of the "marital-status" feature from their dataset. Even if other clients have the same feature, it remains essential for this client to remove any information related to their marital status. This underscores the significance of offering feature unlearning to address individual privacy concerns.
>
> Furthermore, as demonstrated by the experimental results in Tabs 2 and 3 (main paper), our proposed approach, Ferrari, can successfully remove a specific feature for one client even in an **IID setting**. This capability highlights Ferrari's effectiveness in addressing individual feature unlearning requests without compromising the global model's integrity.
>
> Beyond the IID scenario, we also consider the **Non-IID case**. One such example is addressed in response to your fourth question. Additionally, we explore the unlearning backdoor scenario, where backdoor data from one client has a different distribution compared to normal data. This further illustrates the necessity of feature unlearning to ensure model integrity and security in diverse data distributions.
>
> Overall, while data erasure provides a high level of user safety, feature unlearning offers a tailored approach to meet individual privacy needs, making it a crucial aspect of Federated Learning.
>
> `` 2. Proof ``
>
> According to [1], exact feature unlearning can be similarly defined as: $\theta_u \sim \theta_r$, where $\theta_u$ is the unlearned model and $\theta_r$ is the retrained model using data $x+\delta_{\mathcal{F}}$. The proposed Ferrari framework aims to optimize
>
> \begin{equation}
> \min_\theta \mathbb{E}_{(x,y) \in \mathcal{D}_u} \frac{1}{N} \sum\_{i=1}^N \frac{\|\|f\_\theta(x) - f\_\theta (x + \delta\_{F,i})\|\|_2}{\|\|\delta\_{F,i}\|\|_2}
> \end{equation}
>
> This objective leads the unlearned model to approach the retrained model. In particular, when this loss approaches zero, the proposed Ferrari framework achieves exact unlearning.
>
> **Ref:**
>
> #### _[1] Bourtoule et. al, “Machine unlearning,” in 2021 IEEE Symposium on Security and Privacy (SP), pp. 141–159, 2021_
>
> `` 3. Utility Loss ``
>
> Our method is expected to outperform the "naive retrain" method in sensitive feature unlearning scenario. This is because adding Gaussian noise to the target unlearn feature region during retraining reduces model utility, as shown in Fig 1 (main paper). Equation (9) theoretically proves that our framework results in less utility loss than naive retraining, since larger loss leads to lower model performance.
>
> `` 4. Non-IID ``
>
> Our proposed framework remains effective for feature unlearning in Non-IID scenarios, as illustrated in Fig 3 of the rebuttal PDF. The results show that the Ferrari significantly improves feature unlearning performance, with only a 0.2\% drop in $D_u$ when $\gamma = 1$ compared to the IID scenario. Additionally, it maintains $D_r$
> accuracy with only a slight 2\% decrease compared to the Retrain method in the Non-IID scenario.
>
> `` 5. MIA Privacy ``
>
> Ferrari effectively defends against Model Inversion Attacks (MIA) [1, 2, 3] by preventing the reconstruction of the target unlearn features. As shown in Tab 3 (main paper), the attack success rate with our framework is similar to that of the naive retrain method, indicating that unlearned features are protected. In contrast, the baseline model before unlearning has a 90\% attack success rate, highlighting severe data privacy risks. For other baselines like FedCDP and FedRecovery, the success rate is about 80\%. Ferrari reduces this rate to around 50\%, significantly enhancing privacy. Tab 4 (main paper) shows that our framework prevents reconstruction of the target unlearned feature (e.g., the mouth), whereas the baseline model does not.
>
> **Ref:**
>
> ##### [1] _M. Fredrikson et al, “Privacy in pharmacogenetics: An end-to-end case study of personalized warfarin dosing,” Proceedings of the USENIX Security Symposium. UNIX Security Symposium, vol. 2014, pp. 17–32, 2014_
>
> ##### [2] _Fredrikson et al, “Model inversion attacks that exploit confidence information and basic countermeasures,” CCS ’15, (New York, NY, USA), p. 1322–1333, Association for Computing Machinery, 2015_
>
> ##### [3] _Y. Zhang et al, “The secret revealer: Generative model inversion attacks against deep neural networks,” in 2020 IEEE/CVF Conference on Computer Vision and Pattern Recognition (CVPR), pp. 250–258, 2020_
>
> `` 6. MIA Metrics ``
>
> Thank you for your enquiry. First, Model Inversion Attack (MIA) reconstructs the data from the unlearned model. We use the attack success rate of MIA to indicate whether the reconstructed data exposes the feature, as used in [1, 2, 3]. Second, besides MIA evaluation, we also leverage backdoor accuracy, Feature Sensitivity, and visualization tools (GradCAM) to evaluate the unlearning effectiveness of different methods in Section 5.1 (main paper).
>
> The references are the same as the references in question 5 above.
>
> `` 7. Baselines ``
>
> Our study compares against benchmark baselines in federated unlearning, such as FedCDP for class unlearning and FedRecovery for client unlearning, due to the absence of comparable baselines for feature unlearning in Federated Learning. This highlights the novelty and significance of our work. Our results show that current methods like FedCDP and FedRecovery are ineffective for specific feature unlearning scenarios, emphasizing our study's contribution.

---

> > ### Comment · Reviewer_vD3D · 2024-08-10
> > **Thanks for Authors' Rebuttal**
> >
> > Thank you to the authors for their detailed response. I have read the authors' rebuttal carefully and it addresses most of my concerns. Therefore, I will revise my rating appropriately. Thank you again to the authors for their response.

---

> ### Author Response · Authors · 2024-08-11
> **Thank You for Rating Upgrade**
>
> Thank you for your thoughtful feedback. We are pleased to hear that our rebuttal has addressed most of your concerns. We are truly grateful for your decision to revise and improve the rating based on our clarifications. Your willingness to acknowledge the revisions means a great deal to us, and we are committed to addressing any further concerns you may have. We are more than happy to engage in any additional dialogue necessary to ensure that our paper meets the highest standards of quality.
>
> Once again, thank you for your invaluable insights and for your thoughtful consideration in upgrading the score.

---

### Official Review · Reviewer_HLpG · 2024-07-15

**Soundness:** 3
**Presentation:** 3
**Contribution:** 3
**Rating:** 6
**Confidence:** 4

**Summary:**

The paper identifies two main challenges, which is 1) evaluating the unlearning effectiveness without rebuilding the dataset without the unlearned feature, and 2) achieving feature unlearning in Federated Learning without requiring data or computational resources from other clients. The authors define a concept of  ‘feature sensitivity’ based on Lipschitz continuity to evaluate the effectiveness of feature unlearning. This metric measures how sensitive the model output is to perturbations in the input feature. Federated Feature Unlearning (Ferrari), has an objective to minimize ‘feature sensitivity’ to achieve feature unlearning in federated unlearning. It allows clients to unlearn specific features from the global model without needing other clients' participation. The paper provides theoretical proofs and experimental results demonstrating effectiveness in various scenarios (sensitive, backdoor, biased feature unlearning). It shows minimal loss in model utility compared to exact feature unlearning methods.

**Strengths:**

1.  Novelty: The presented framework, Ferrairi, appers to be novel, as the focus on minimizing feature sensitivity is a new perspective on ensuring data privacy and integrity in federated unlearning.

2.  Effectiveness: It is a practical and effective solution for federated unlearning, without requiring participation of all clients. It is designed to require only the local datasets from the unlearned clients, thereby reducing the overhead. It also achieves effectiveness by achieving unlearning in a short amount of time.

3. Extensive experiments, and theoretical analysis and proof: The authors provide empirical experimental results through various datasets and scenarios, while showing the effectiveness of their methods. Also, their theoretial analysis shows that ferrari achives less utility loss compared to exact feature unlearning, reinforcing its effectiveness.

**Weaknesses:**

1. There is no evaulation on the scalability of the proposed framework. While the experiments used various datasets, the scale and dataset sizes seem to be relatively small. (MNIST, CIFAR, CelebA). It would be better to provide results on performance on large scale environments to prove generalization ability.

2. While it is true that the suggested Ferrari has significant reduction in terms of unlearning time, in terms of computational complexity, it would be better to include actual computational cost metrics such as FLOPs.

3.  The theoretical analysis in the paper relies on certain assumptions, such as zero training loss and specific data distribution characteristics, which may not always hold in real-world scenarios.

4. Also there are a few minor mistakes and typos in the paper.

**Questions:**

1. Can you provide more insights into the computational overhead and scalability of Ferrari, particularly when dealing with a lot of clients?

2. In the experiments, learning rate and Gaussian noise levels are mensioned as hyperparmeters. How much is the Ferrari sensitive to hyperparameters like these, or other than these?

**Limitations:**

1. In federated learning, clients may drop out or join during training or unlearning. Or the performance effect of using federated unlearning on long term is unknown. I would like to suggest about considering making the framework robust to these kinds of real world scenarios.

---

> ### Author Rebuttal · Authors · 2024-08-07
>
> `` 1. Large-Scale Dataset ``
>
> Thank you. As requested, we conducted additional experiments using the ImageNet dataset to assess the generalization ability of our proposed framework. Below are the tables summarizing the results. These results demonstrate that our framework performs well  even in large-scale environments, confirming its robustness and ability to generalize effectively. Specifically, the results show low $D_u$ accuracy while maintaining high test accuracy, indicating high model utility.
>
> > * Model Utility Analysis:
>
> | **Scenarios** | **Datasets** | **Unlearn Feature** | **Baseline**       | **Retrain**        | **Fine-tune**      | **FedCDP**         | **FedRecovery**    | **Ours** |
> |---------------|--------------|---------------------|--------------------|--------------------|--------------------|--------------------|--------------------|------------------------|
> | Backdoor  | ImageNet | Backdoor Pixel Pattern | 52.86 ± 4.14      | 67.18 ± 2.07       | 67.52 ± 1.69| 31.17 ± 3.96       | 12.75 ± 5.27       | 65.36 ± 1.84           |
>
> > * Unlearning Effectiveness Analysis:
>
> | **Scenarios** | **Datasets** | **Unlearn Feature** |       | **Baseline**       | **Retrain**        | **Fine-tune**      | **FedCDP**         | **FedRecovery**    | **Ours**           |
> |---------------|--------------|---------------------|-------|--------------------|--------------------|--------------------|--------------------|--------------------|--------------------|
> | Backdoor  | ImageNet | Backdoor Pixel Pattern | **$\mathcal{D}_r$** | 52.35 ± 2.25       | 67.05 ± 1.29       |67.34 ± 2.73| 29.74 ± 4.72       | 13.46 ± 6.53       | 65.74 ± 1.32       |
> |               |              |                     | **$\mathcal{D}_u$** | 83.16 ± 3.74       | 0.00 ± 0.00        | 71.48 ± 3.69       | 62.39 ± 3.05       | 54.92 ± 5.59       | 0.09 ± 0.02    |
>
> `` 2. Computational Overhead & Scability ``
>
> Thank you. As requested, We present the computational complexity of Ferrari using FLOPs metrics, as shown in Fig 5 of the rebuttal PDF. Our Ferrari framework exhibits the lowest FLOPs, achieved by accessing only the local dataset of the unlearned client and minimizing feature sensitivity within a single epoch.
>
> For scalability, Ferrari remains effective in large-scale FL environments, as shown in the client number analysis in Fig 4 of the rebuttal PDF. It exhibits less than 0.5\% accuracy deterioration for the retain client dataset $D_r$ and less than 0.1\% for the unlearn client dataset $D_u$.
>
> `` 3. Theoretical Analysis ``
>
> Thank you and we will elaborate more on Assumption 1-2.
>
> - First, Assumption 1 elucidates that the utility loss associated with a perturbation norm less than $C $ is smaller than the utility loss when the perturbation norm is greater than $C$. **This assumption is logical, as larger perturbations would naturally lead to greater utility loss.**
>
> - Second, **we could also extend the Theorem 1 to the case of the non-zero training loss**. This is because
>
> \begin{equation}
>    \mathbb{E}\_{(x,y)\in \mathcal{D}} \left( \|\| f\_{\theta^*}(x) - y \|\| + \lambda \mathbb{E}\_{\|\| \delta_F \|\| \geq \lambda} \frac{\|\|f\_\theta(x) - f\_{\theta^*} (x + \delta\_{F})\|\|_2}{\|\|\delta\_{F}\|\|_2} \right) \\
> \end{equation}
>
> \begin{equation}
>   = \mathbb{E}_{(x,y)\in \mathcal{D}} \mathcal{l}\|\| f\_{\theta^*}(x) - y \|\| + \lambda \mathbb{E}\_{(x,y)\in \mathcal{D}} \mathbb{E}\_{\|\| \delta_F \|\| \geq \lambda} \frac{\|\|f\_\theta(x) - f\_{\theta^*} (x + \delta\_{F})\|\|_2}{\|\|\delta\_{F}\|\|_2}
> \end{equation}
>
> \begin{equation}
>   \leq \mathbb{E}_{(x,y)\in \mathcal{D}} \mathcal{l} \|\| f\_{\theta^*}(x) - y \|\| + \mathbb{E}\_{(x,y)\in \mathcal{D}} \mathbb{E}\_{\|\| \delta_F \|\| \geq \lambda} \|\|f\_\theta(x) - f\_{\theta^*} (x + \delta\_{F})\|\|_2
> \end{equation}
>
> \begin{equation}
>   \leq  \mathbb{E}\_{(x,y)\in \mathcal{D}} \mathbb{E}\_{\|\| \delta_F \|\| \geq \lambda} \|\|y - f\_{\theta^*} (x + \delta\_{F})\|\|_2
> \end{equation}
>
> which obtain Eq. (13) in appendix (noted that $\|\delta_\mathcal{F}\| \geq \frac{1}{\lambda}$ in original appendix should be $\|\delta_\mathcal{F}\| \geq {\lambda}$).
>
> `` 4. LR & Gaussian Noise ``
>
> We considered the sensitivity of Gaussian noise levels in our proposed framework, as shown in Fig 7(a) (main paper).  Within the range from 0.05 to 1.0, retain client dataset $D_r$ accuracy stays high and unlearn client dataset $D_u$ accuracy remains low, indicating a balance. Thus, we implement noise value within this range in our proposed framework for balanced accuracy across $D_r$ and $D_u$.
>
> In Fig 2 of the rebuttal PDF, we analyze the learning rate hyper-parameter for Ferrari, ranging from 0.00001 to 0.01. Our findings show that a larger learning rate slightly decreases accuracy on both $D_r$ and $D_u$, while a lower learning rate slightly improves accuracy. A learning rate of 0.0001 provides a satisfactory balance, with less than a 2\% change in accuracy for $D_r$ and less than a 0.2\% change for $D_u$. Thus, we selected 0.0001 for all experiments in this study.
>
> Therefore, the unlearning effectiveness of our proposed framework is not sensitive to the learning rate but is sensitive to the level of Gaussian noise.
>
> `` 5. Client Participation Affect ``
>
> Thank you for your suggestion. We would like to clarify that our proposed framework remains effective even when clients drop out or join during training or unlearning. This robustness is due to our framework’s reliance solely on the participation of the unlearning client and their local dataset for local feature unlearning. As demonstrated in Fig 2 and Alg 1 of our main paper, our framework minimizes feature sensitivity, ensuring effective unlearning regardless of broader client participation. **This adaptability is a key advantage of our proposal.**
>
> ``6. Minor Mistake``
>
> Thank you for pointing out the minor mistakes and typos in the paper. We have carefully reviewed the manuscript and corrected these issues.

---

> > ### Comment · Reviewer_HLpG · 2024-08-14
> >
> > Thanks for the rebuttals and providing the responses. They mostly clarified my concerns. So, I will increase my rating.

---

### Author Rebuttal · Authors · 2024-08-07

Dear all reviewers,

Thank you for your valuable comments and thoughtful inquiries regarding our work. We appreciate the opportunity to address your concerns and are committed to providing further clarity.

To better illustrate our responses, we have uploaded a PDF document containing new figures in this section. We have also included detailed explanations and clarifications in the respective rebuttal sections for each reviewer.

We hope these responses adequately address your concerns. We are committed to making the necessary improvements to our revised manuscript.

Thank you again for your feedback.

---

### Decision · Program_Chairs · 2024-09-25

**Decision:**

Accept (poster)

**Comment:**

This paper proposes a Lipschitz-based feature unlearning measure and a Lipschitz-based method to perform feature unlearning in the federated setting with only one client's participation.
The paper shows that the loss in utility for feature unlearning is small using their approach and can handle various feature unlearning scenarios such as sensitive, backdoor and biased features.

After some discussion, the reviewers agreed that this approach provides a novel perspective on the problem. The experiments in the original paper and the author response provide the empirical support for the effectiveness of the feature unlearning.

The authors should include additional discussion of their theoretic assumptions. These are critical for understanding the context of the theoretic results. For example, I am not sure that assumption 1 is always reasonable. For example, suppose the perturbation makes all the pixels equal to the mean of the original patch---this is similar to Fig 1(c). This is probably a smaller perturbation than Gaussian noise as in Fig 1(b). Yet, the performance is better with a "larger" perturbation in this case. Thus, I think the "kind" of perturbation matters significantly due to the inductive bias of the NN architecture and basic intuition does not necessarily hold here. Nonetheless, I think it is still a reasonable assumption but should not be taken as obvious in all cases and more nuanced discussion is needed.

It also should be clarified that Ferrari is better than "naive" retraining without the feature, but may not be better than a non-naive retraining method. Also, the difference between retraining with IID vs non-IID should be discussed; there does seem to be a gap, albeit relatively small at 2% but this should be noted. These should be clarified to avoid overclaiming in the paper.

Overall, the paper presents a novel federated feature unlearning method using Lipschitz continuity metric and regularization to reduce feature sensitivity while retaining accuracy close to naive retraining.